# Experimental demonstration of a skyrmion-enhanced strain-mediated physical reservoir computing system

Yiming Sun [1,12], Tao Lin[1,12], Na Lei [1,12] ✉, Xing Chen[1,12], Wang Kang[1], Zhiyuan Zhao[2], Dahai Wei[2], Chao Chen[1], Simin Pang[2,3], Linglong Hu[4], Liu Yang[1], Enxuan Dong[1], Li Zhao[5], Lei Liu [2], Zhe Yuan [5], Aladin Ullrich [6], Christian H. Back [7,8,9], Jun Zhang [2,3,10], Dong Pan [2], Jianhua Zhao[2], Ming Feng [4] ✉, Albert Fert [1,11] & Weisheng Zhao [1]

Physical reservoirs holding intrinsic nonlinearity, high dimensionality, and memory effects have attracted considerable interest regarding solving complex tasks efficiently. Particularly, spintronic and strain-mediated electronic physical reservoirs are appealing due to their high speed, multi-parameter fusion and low power consumption. Here, we experimentally realize a skyrmion-enhanced strain-mediated physical reservoir in a multiferroic heterostructure of Pt/Co/Gd multilayers on (001)-oriented 0.7PbMg$_{1/3}$Nb$_{2/3}$O$_3$–0.3PbTiO$_3$ (PMN-PT). The enhancement is coming from the fusion of magnetic skyrmions and electro resistivity tuned by strain simultaneously. The functionality of the strain-mediated RC system is successfully achieved via a sequential waveform classification task with the recognition rate of 99.3% for the last waveform, and a Mackey-Glass time series prediction task with normalized root mean square error (NRMSE) of 0.2 for a 20-step prediction. Our work lays the foundations for low-power neuromorphic computing systems with magneto-electro-ferroelastic tunability, representing a further step towards developing future strain-mediated spintronic applications.

Reservoir computing (RC) is a computational framework of recurrent neural networks (RNNs) in neuromorphic computing, suited to temporal/sequential information processing[1–3]. The reservoir is a network of recurrently and randomly connected nonlinear nodes, where input data are mapped into a high-dimensional space and become linearly separable at the output nodes. In particular, all the recurrent connections inside the reservoir are fixed, and only the external connections (between the reservoir and an output layer), $W_{out}$, need to be trained with a simple method, such as linear regression. The training procedure is greatly simplified, making the learning rapid and stable.

[1]Fert Beijing Institute, MIIT Key Laboratory of Spintronics, School of Integrated Circuit Science and Engineering, Beihang University, Beijing 100191, China. [2]State Key Laboratory for Superlattices and Microstructures, Institute of Semiconductors, Chinese Academy of Sciences, Beijing 100083, China. [3]Center of Materials Science and Optoelectronics Engineering, University of Chinese Academy of Sciences, Beijing 100049, China. [4]Key Laboratory of Functional Materials Physics and Chemistry of the Ministry of Education, Jilin Normal University, Changchun 130103, China. [5]The Center for Advanced Quantum Studies and Department of Physics, Beijing Normal University, Beijing 100875, China. [6]Institute of Physics, University of Augsburg, Augsburg 86159, Germany. [7]Department of Physics, Technical University of Munich, Garching 85748, Germany. [8]Munich Center for Quantum Science and Technology (MCQST), Munich 80799, Germany. [9]Centre for Quantum Engineering (ZQE), Technical University of Munich, 85748 Garching, Germany. [10]CAS Center of Excellence in Topological Quantum Computation, University of Chinese Academy of Sciences, Beijing 100049, China. [11]Unité Mixte de Physique, CNRS, Thales, Université Paris-Saclay, Palaiseau 91767, France. [12]These authors contributed equally: Yiming Sun, Tao Lin, Na Lei, Xing Chen. ✉e-mail: na.lei@buaa.edu.cn; mingfeng@jlnu.edu.cn

Therefore, the computational cost of a RC system is significantly reduced compared to typical RNNs.

In recent years, worldwide interest has been gained in the physical implementation of RC systems, in which the reservoirs can accelerate data processing and reduce the learning cost. The physical reservoirs (systems/substrates/devices) should possess intrinsic high dimensionality, nonlinearity, and short-term memory characteristics[4–27]. The complex nonlinear dynamics guarantee energy-efficient machine learning, while the short-term memory effect is crucial for temporal/sequential processing where the history of the input signal is influential. A rich variety of physical reservoirs has been proposed and demonstrated, including electronic[4–8], photonic[9–11], mechanical[12], spintronic[13–27] reservoirs, etc. Among these, spintronic reservoirs are appealing for scalable and low-power physical implementations of RC systems due to their non-volatility, nonlinear dynamics, multi-functionality, and complementary metal–oxide–semiconductor (CMOS) compatibility[15,16].

To date, spin-torque nano-oscillators[13,14], spin waves[17], dipole-coupled nanomagnets[18], and magnetic skyrmions[19–27] have been proposed for realizing physical reservoirs. Magnetic skyrmions exhibiting small size (sub-10 nm), high energy efficiency, and especially topological stability[28–42] show exclusive advantages for implementing a reservoir regarding nonlinear response, memory of past manipulations and complex interaction between multiple skyrmions. Thus, characters of skyrmion displacement and deformation are utilized to propose RC in individual skyrmion and skyrmion fabrics systems[19–22]. Recently, skyrmion-based reservoirs have been successfully demonstrated to realize the benchmark tasks including Boolean logic operations, pattern recognition, and chaotic time series forecasting[23–25]. However, either the electrical currents or magnetic fields are served as inputs, which is not advantageous to lower energy costs. Accordingly, a strain-mediated voltage-controlled skyrmion RC block employing the nonlinear breathing dynamics has been proposed, with an estimated energy dissipation of 50 fJ per single input, which is three-four orders of magnitude lower than that of the CMOS-based reservoir[26]. Moreover, strain is a universal way to control various characteristics, which have been widely studied, including magnetization[43–48], resistance[49,50], Dzyaloshinskii–Moriya interaction (DMI)[51], phase transitions[52], luminescence[53], etc. Hence, multiferroic heterostructures inherently capable for multi-parameter fusion are promising to be a powerful physical reservoir for different tasks. Here, we proposed a strain-mediated Hall bar device using the electric-field (E-field) as inputs and the anomalous Hall effect (AHE) response as outputs that are all electrical voltage signals. We experimentally realize a skyrmion-enhanced strain-mediated physical reservoir by combining the magnetization and resistivity changes. Furthermore, we demonstrate its functionality via a sequential waveform classification task and a Mackey-Glass time series prediction task. Our work opens a new route for low-power neuromorphic computing.

Figure 1 illustrates the framework of our skyrmion-enhanced strain-mediated RC system. A Hall bar device fabricated on a PMN-PT(001) substrate functions as a physical reservoir, where a voltage sequence V(t) is applied across the substrate, serving as the input, with the Hall voltage $V_{xy}$ as the output signal. The bottom panel of Fig. 1 illustrates that E-field induced compressive in-plane strain, which could change both the magnetization $M_z$ and the longitudinal resistivity $\rho_{xx}$.

The empirical relationship between the Hall resistivity $\rho_{xy}$[54] and the applied perpendicular magnetic field $H_z$, $M_z$, and $\rho_{xx}$[55,56] is modified as follows:

$$\rho_{xy}(E) = R_0 H_z + R_s 4\pi M_z(E) + a\rho_{xx}(E) \qquad (1)$$

where E is the applied E-field across the PMN-PT substrate and $R_0$, $R_s$, and $a$ are the coefficients for the ordinary Hall, the anomalous Hall, and

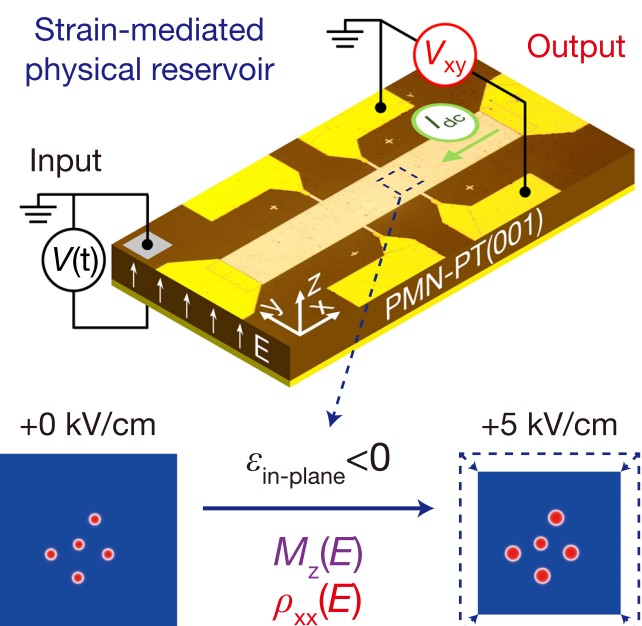

**Fig. 1 | Schematic diagrams of the experimental method and the skyrmion-enhanced strain-mediated spintronic RC system.** The top panel shows the AHE measurement setup under a voltage sequence V(t), where the top and bottom surfaces are ground and voltage applied, respectively. The central light-yellow rectangle is the Hall bar whose size is $150 \times 900\ \mu m^2$. The physical reservoir is realized using the strain-mediated spintronic AHE device, which is confirmed as a nonlinear dynamic system. The Hall voltage $V_{xy}$ is the output signal of this system. The yellow parts are Au electrodes for wire bonding and the grey part is conductive silver paint to minimize contact resistance. The bottom panel illustrates that the $M_z$ (skyrmion size change) and $\rho_{xx}$ both varies with the applied E-field due to the inverse piezoelectric effect, where in-plane compressive strain $\varepsilon_{in-plane} < 0$ is generated.

the longitudinal resistivity, respectively. The applied E-fields generate the ferroelastic strains and further tune the magnetization and resistivity of the magnetic layers.

## Results
### Characterization of the nonlinearities
The nonlinearity of this skyrmion-enhanced strain-mediated physical reservoir is examined, where the magnetization and resistivity variation behaviours are measured independently from the applied voltages across the PMN-PT(001) substrate. First, the magnetization changes under the application of an E-field are studied using the magneto-optical Kerr effect (MOKE), and the corresponding hysteresis loops are shown in Fig. 2a. Cross-sectional transmission electron microscopy (TEM) is performed to confirm the quality of the interfaces (see supplementary information S1) and the magnetic states in the Pt/Co/Gd multilayers are examined by magnetic force microscopy (MFM) under different magnetic fields and voltages. The transition of the magnetic states from labyrinth domains to skyrmions and finally to the saturation state when increasing the magnetic field are further confirmed by Lorentz transmission electron microscopy (L-TEM), for details see supplementary information S2. When the E-field is applied, the magnetization nucleation fields increase significantly from 100 to 150 mT at E = 20 kV/cm, indicating that the skyrmion states are modified by the applied E-fields. As shown in the upper insets of Fig. 2a, MFM images are taken under the magnetic field $\mu_0 H_a$ = 90 mT with no voltage applied (I) and at 10 kV/cm (II). Upon applying and increasing the E-field, skyrmions start to nucleate, and a low-density skyrmion phase (I) evolves to a high-density skyrmion phase (II). To investigate the relationship of the magnetization with the E-fields, we extract the remnant magnetization $M/M_s$ at $\mu_0 H_a$ = 200 and 90 mT, where $M_s$

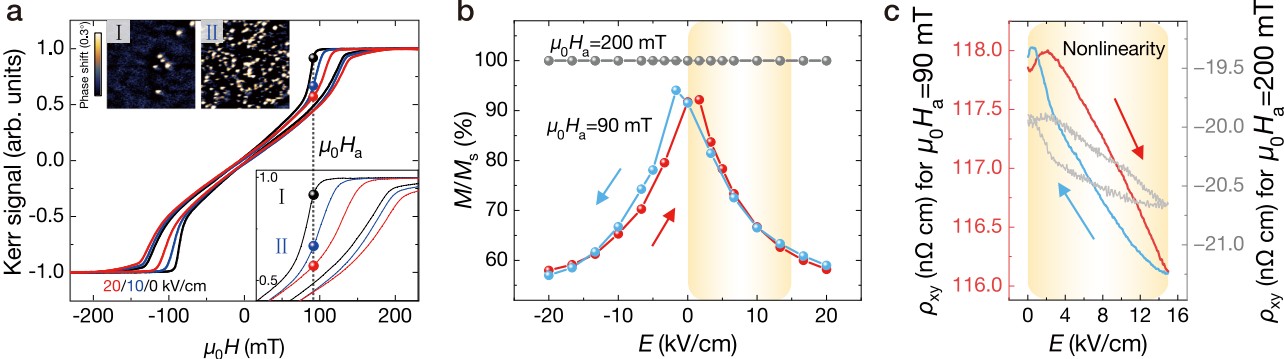

**Fig. 2 | Evaluation of the *E*-field effects and nonlinearity of the reservoir. a** Out-of-plane magnetic hysteresis loops under different *E*-fields. The bottom inset shows the enlarged loops around $\mu_0 H_a$, where the skyrmions start to nucleate. The MFM phase shift images of stages I and II are shown as insets. The image size is $5 \times 5 \, \mu m^2$. **b** $M/M_s$ extracted from **a** versus the *E*-field. The arrows indicate the *E*-field scanning directions. **c** Nonlinearity of the Hall resistivity $\rho_{xy}$ versus the *E*-field, where $\mu_0 H_a = 90$ mT (for the red and blue curves) and 200 mT (for the grey curve).

denotes the saturated magnetization, as shown in Fig. 2b, corresponding to the saturation and skyrmion phases. Through counting the skyrmion area from the MFM images in the Fig. 2a insets I and II, $\Delta M_{skyrmion}/M_s$ are 1.3% and 13%, respectively (see supplementary information S3 for detailed results and analysis). $\Delta M(E)$ for *E*-fields between 0 and 10 kV/cm is 11.7%, which is in a good agreement with the results shown in Fig. 2b, where the magnetization change is about 13%. We can conclude that the magnetization decrease comes from the skyrmion density increase with *E*-field applied. Further extracting the magnetization change versus *E*-fields, a typical butterfly shape is shown apparently at $\mu_0 H_a = 90$ mT, indicating the strain-mediated modulation of the skyrmion phase realized in this heterostructure. Note that, the skyrmion phase change mainly stems from the strain induced magnetic anisotropy change, that exhibits a similar butterfly shape (see supplementary information S4 for detailed results and analysis). Such nonlinearity of the magnetization originates from the nonlinear variation of the piezoelectric strain versus the applied *E*-field with ferroelectric switching fields of ±1.67 kV/cm[57,58]. Therefore, the skyrmion phase controlled by an *E*-field is preferable for the RC application to achieve greater nonlinearity in comparison to the saturation state in multiferroic heterostructures.

Second, the variation of the Hall resistivity $\rho_{xy}$ with the applied *E*-fields is measured, as shown in Fig. 2c. By scanning the unipolar *E*-field from 0 to 15 kV/cm and then back to 0 in a time period of 50 s (the red and blue arrows indicate the scan directions) at $\mu_0 H_a = 90$ mT (the skyrmion state), $\rho_{xy}$ shows a half branch of a typical butterfly shape from piezoelectric strain with partial contribution of 109° ferroelastic domain switching[48,49]. When sweeping the *E*-field from zero, $\rho_{xy}$ first increases slightly to the switching *E*-field and then decreases sharply. When sweeping the *E*-field back, a typical hysteresis increase of $\rho_{xy}$ can be seen, and the curves cross over the first half cycle (the red curve) when the *E*-field is close to zero, which induces a slight positive offset at 0 kV/cm. The grey curve in Fig. 2c shows the nonlinear $\rho_{xy}$ variation at $\mu_0 H_a = 200$ mT (the saturation state), where only the change of longitudinal resistivity with *E*-field occurs. Thus, the amplitude is smaller than the one in the skyrmion state, where an additional change of the magnetization combines with the longitudinal resistivity change. In the working regime for the sequential waveform classification task (marked by the yellow shaded area in Fig. 2b, c), i.e. the positive *E*-field range, both the Hall resistivity $\rho_{xy}$ and the magnetization in the skyrmion state behave nonlinearly with the input voltages, implying that a better performance of the strain-mediated RC system can be expected compared to the saturated magnetization state. We can conclude that the strain-control of skyrmions in PMN-PT/Pt/Co/Gd multilayers would be a capable candidate for a physical reservoir with rich nonlinearity.

## Sequential waveform classification task

In this part, we use a representative sequential waveform classification task[21,24,26] to test the performance of our skyrmion-enhanced strain-mediated RC system. The input signal is a random waveform sequence comprising square and sine waveforms (labelled as '0' and '1', respectively) encoded as the voltage $V(t)$ applied to the RC system, and the output signal is the AHE voltage $V_{xy}$.

The AHE measurement system is shown in Fig. 1, and a constant current $I_{dc} = 0.5$ mA ($j_{dc} = 9.8 \times 10^3$ A/cm²) is applied along the *x*-axis of the Hall bar device, where the Joule heating and magnetic structure changing are negligible due to the extremely low current density. Figure 3a illustrates the schematic of the strain-mediated spintronic RC framework, where the two different magnetization states (skyrmion phase and saturation) serve as different physical reservoirs, presented by MFM images. Figure 3b shows a segment of randomly extracted input signals (black waves on the top panel) and the corresponding output signals from the strain-mediated spintronic reservoir. The red and grey output signals correspond to the skyrmion and saturation states, respectively, as shown in the bottom panel of Fig. 3b. Note that the output signals are offset to zero, centred to eliminate the background voltages caused by different magnetic fields. The *E*-field-induced anomalous Hall voltage change $\Delta V_{xy}$ of the skyrmion state reservoir is around 3 μV (red output signal), while the saturation (grey output signal) shows much smaller amplitudes around 1.5 μV as displayed in Fig. 3b. The behaviour of the output signal coincides with the nonlinearity shown in Fig. 2c, and obvious differences can be seen around 0 kV/cm with single and double peaks for square and sine waveforms, respectively. The gradually approaching to and departing from 0 kV/cm in the input waveform both induce peaks in the output signals, but the nonlinearity is less significant around 15 kV/cm. Relatively, the memory effect in our systems can be derived from the gradual decrease/increase of the output signals around 15 kV/cm after a series of square/sine waveforms, which is consistent with the previous reported phenomena[59,60].

The purpose of the waveform recognition task is to recognize the current (black square), last (green square), and second-to-last (orange square) waveform types (Fig. 3c) by using the sampled *n* data points from the current period of output signal and the trained weights corresponding to each point (see methods). The waveform sequences '0 1 1' and '0 0 1' are marked by the yellow and blue regions. The last cycle of the input signals in these two regions are different (sine wave in the yellow region and square wave in the blue region). The orange, green, black squares, and the yellow dots correspond to those in Fig. 3c. The coloured lines connecting squares and dots represent the output weight matrix $W_{out}$. The dotted and solid curves in Fig. 3d show

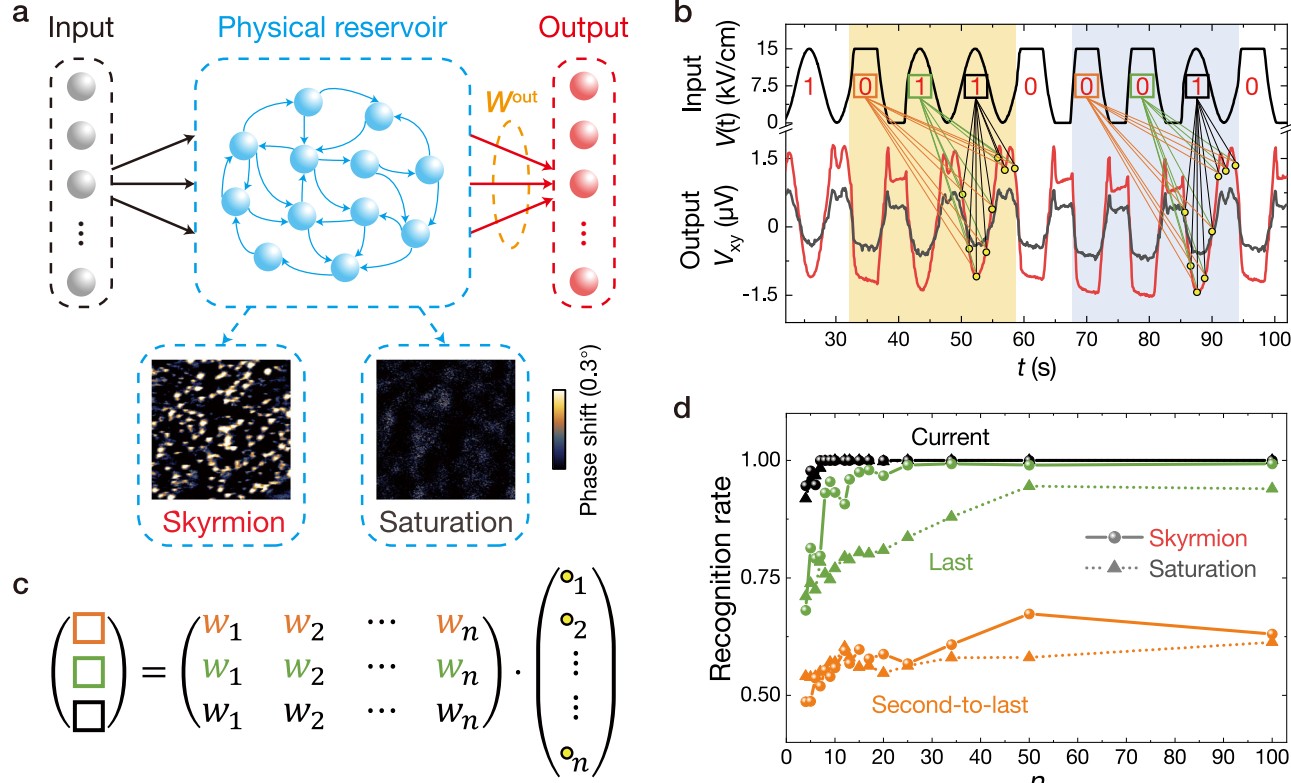

**Fig. 3 | Sequential waveform classification task via the strain-mediated spintronic RC system. a** Schematic of the strain-mediated spintronic RC system. Two different magnetization states are studied, as shown in the MFM phase shift images. The image size is $5 \times 5 \, \mu m^2$. **b** Input waveform sequences and the corresponding output signals of the strain-mediated spintronic reservoir. The red and grey output signals correspond to the skyrmion and saturation states, respectively. The output signal is sampled by $n$ data points per period, marked by the yellow dots. **c** Output values are summed by weight for three different sets of weights. One set is trained to recognize the current waveform (black), one is trained to recognize the last waveform (green) preceding the current waveform, and the other is trained to recognize the second-to-last waveform (orange). **d** Recognition rates as a function of $n$ for two magnetization states (skyrmion and saturation states) of the strain-mediated spintronic RC system.

the recognition rates for the saturation and skyrmion state-based strain-mediated RC systems, respectively. The black, green, and orange curves in Fig. 3d show the recognition rates of the current, last, and second-to-last waveforms as a function of $n$, the number of data points sampled per period at $V_{xy}$. For the current waveform, the recognition rates reach 100% (perfectly classified) on increasing $n$. The system will provide better efficiency to run a reservoir computer if a smaller $n$ is used to realize higher accuracy in a task. For the second-to-last waveform, the recognition rates of both RC systems are all around 50%, indicating randomness and unpredictability. Interestingly, the two RC systems have similar trends with $n$; however, distinctive differences are observed for the last waveform recognition. Upon increasing $n$, the recognition rates increase rapidly with obvious randomness and then tend to saturate with $n > 10$. The recognition rates are 94.4% and 99.3% in the saturation and skyrmion states, respectively. The skyrmion-enhanced strain-mediated reservoir shows a better performance regarding the recognition rate for the last waveform. This result demonstrates the short-term memory effect of the strain-mediated reservoir, which is 2-waveform duration time (specifically to our experiment, the time scale is about $9 \, s \times 2 = 18 \, s$). The physical mechanism of the skyrmion enhancement is discussed later (see discussion section and supplementary information S4, S5). The reservoir's computational capabilities are evaluated by further analyzing the experimental data of the waveform classification task[61,62], where $T_{delay, \, max} = 2$ is used in the calculation since the experimental results of the waveform classification only comprise the current, the last, and the second-to-last waveform. The parity check capacity ($C_{PC}$) and short-term memory content ($C_{STM}$) at the skyrmion state are both

around 2.31, which are restricted by the slow operation of the present configuration. However, they could be significantly enhanced through involving the fast magnetization dynamics[26].

## Mackey-Glass time series prediction task

In addition to the task of waveform recognition, our strain-mediated spintronic reservoir can also implement more complex tasks, e.g. Mackey-Glass (MG) time series prediction, a benchmark task for reservoir computing[5,25–27]. In the MG prediction task, the reservoir input corresponds to a MG chaotic time series, which is generated from a delay differential equation (DDE),

$$\frac{dx(t)}{dt} = \frac{\beta x(t - \tau)}{1 + x^{10}(t - \tau)} - \gamma x(t) \qquad (2)$$

where $x(t)$ is a dynamical variable, and we set the parameters $\beta = 0.2$, $\gamma = 0.1$ and $\tau = 17$ to obtain chaotic dynamics[63].

We first construct a model to demonstrate that the system is able to perform this complicated task (see supplementary information S6) for both skyrmion and saturation states. In the ideal case where experimental noise is ignored, the RC systems in both magnetic states exhibit equivalent performance. This suggests that the better performance in waveform recognition in the skyrmion state is mainly due to the higher signal amplitude. Therefore, it is imperative to increase the total output signal amplitude in order to identify the out-of-noise signal induced by small variations of the input signal. By applying a negative polarized $E$-field, we achieve amplitudes of output $V_{xy}$ in tens of mV range with sweeping $E$-field in between 0 and 15 kV/cm, which is

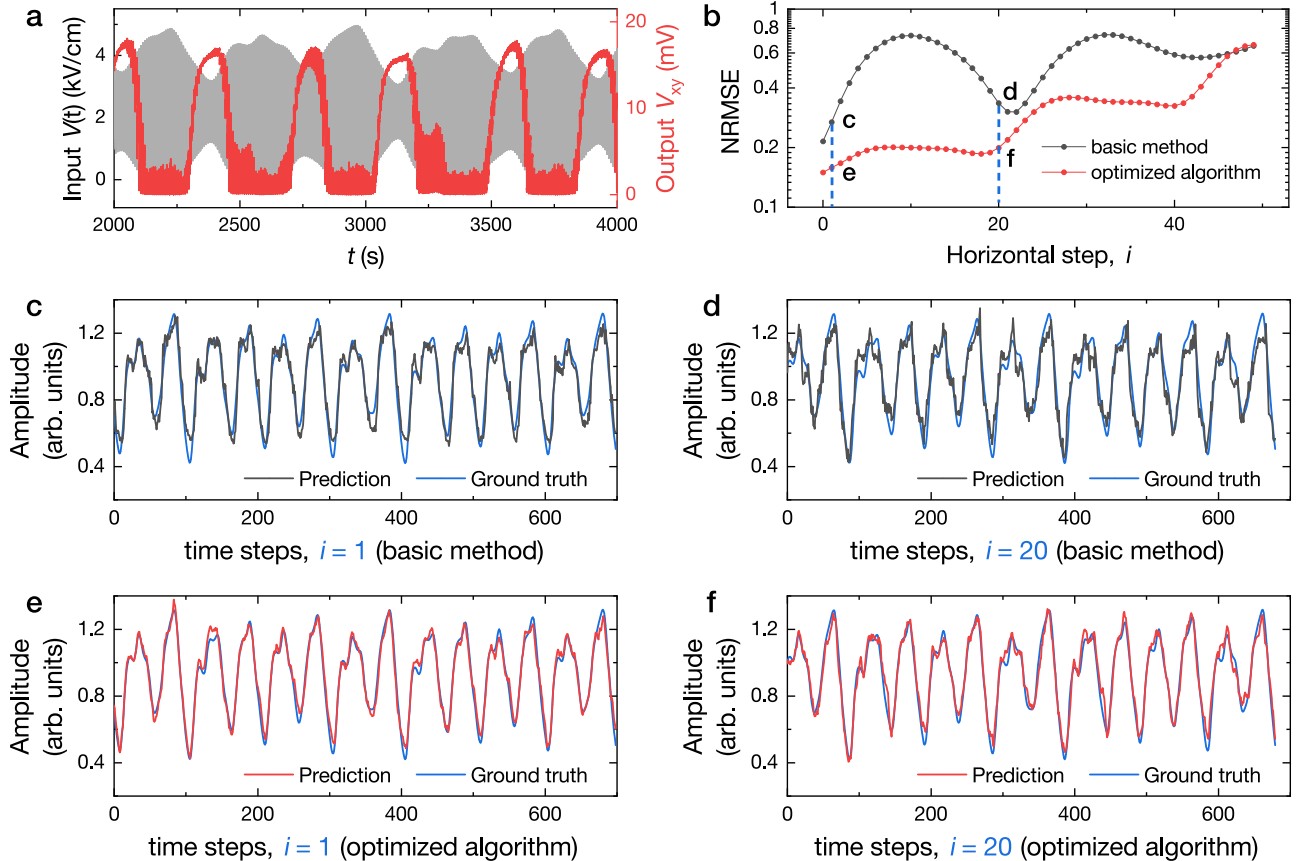

**Fig. 4 | Mackey-Glass time series prediction task demonstrated by the strain-mediated spintronic RC system. a** Input pre-processed signal (grey data set) and the corresponding output signal (red data set) of the strain-mediated reservoir. **b** NRMSE as a function of horizontal prediction step $i$, shown in log scale, for the testing set by using the basic method (black curve) and the optimized algorithm (red curve) for output reconstruction. **c** and **d** are the selected predicting results (black) for horizontal step $i = 1$ (for short-term prediction) and 20 (for long-term prediction), respectively, predicted by the strain-mediated RC system using basic method for output reconstruction, in comparison with the ground truth (blue) of MG time series. **e** and **f** are the selected predicting (red) results for horizontal step $i = 1$ and 20, respectively, predicted by the strain-mediated RC system using optimized algorithm for output reconstruction, in comparison with the ground truth (blue) of MG time series.

the non-volatile longitudinal resistivity change induced by 109° ferroelastic domain switching presumably (see supplementary information S7).

The task of MG time series prediction is then performed experimentally. As shown in Fig. 4a, the physical reservoir receives a pre-processed sequence of electric voltage inputs (the variation range of the $E$-field is in between 0 and +5 kV/cm, the grey data set), and the output is obtained from the Hall voltage $V_{xy}$ (the red data set) multiplied by a trained output matrix $W_{out}$. The number of steps between the future time step and the current step is defined as prediction horizontal step $i$. The matrix $W_{out}$ is different for different $i$ values. The details about the input signal pre-processing and the output matrix $W_{out}$ training procedure are introduced in the methods section.

To maximize the use of the collected reservoir dynamics, a previously proposed method is used for output reconstruction[27]. Instead of solely relying on the reservoir's output at current step for output reconstruction, we can enhance the information available for training through combining the response states at each time step with the output states from the previous time steps. This creates a new state matrix that has a larger state dimensionality of $N_r \cdot (1 + n_p)$, where $n_p$ represents the number of previous steps' states and $N_r$ represents the original reservoir size (see methods). This technique helps to incorporate more information about the reservoir states, effectively increasing the size of the reservoir from $N_r$ to $N_r \cdot (1 + n_p)$.

To evaluate the performance of the trained matrix $W_{out}$, normalized root mean square error (NRMSE) is calculated on the prediction results of the testing set $y_{pre}$ compared to the true trajectory of MG series $y_{tar}$,

$$\text{NRMSE} = \sqrt{\frac{1}{n_s \sigma_{tar}^2} \sum_{i=0}^{n_s} (y_{tar}(i) - y_{pre}(i))^2} \quad (3)$$

A lower value of the NRMSE represents a more accurate prediction. In Fig. 4b, the red and the black curves show the result of NRMSE (in log scale) as a function of horizontal prediction step $i$, using the optimized algorithm ($n_p = 30$) and the basic method (without compensation, i.e. $n_p = 0$) for output reconstruction, respectively. Better accuracy of prediction can be achieved by using the optimized algorithm.

Figure 4c–f shows the predicting results of MG time series for the next value ($i = 1$), and the value happening 20 steps later ($i = 20$), for both the basic method and the optimized algorithm. Each horizontal prediction step $i$ corresponds to a different prediction task. In the figures, the blue curve is the ground truth, the black and red curves are prediction results from basic and optimized methods, respectively. In general, it is more difficult to predict for later steps due to the chaotic nature of the MG time series. The prediction error apparently increases in Fig. 4d compared to 4c, and also the same trend shown in Fig. 4f compared to 4e. In addition, the predicted results using the optimized algorithm shown in Fig. 4e, f demonstrate better accuracy than those using the basic method shown in Fig. 4c, d, respectively. To be specific, the NRMSE values are 0.16 and 0.2 for predictions of $i = 1$ and $i = 20$

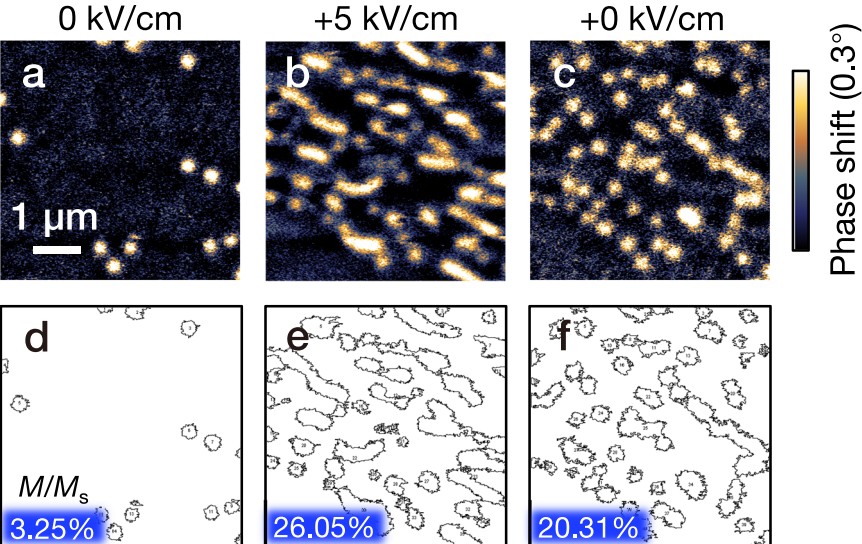

**Fig. 5 | MFM phase shift images under different *E*-fields in the skyrmion state at** $\mu_0 H_a$ **= 90 mT. a** MFM image taken after reducing the magnetic field from saturated state to $\mu_0 H_a$ = 90 mT without applied *E*-field, a few skyrmions generate. **b** MFM image with positive *E*-field of +5 kV/cm applied, a large amount of skyrmions appears with deformed shape. **c** MFM image taken after reducing the *E*-field to +0 kV/cm, skyrmions restore to circular shape. The scale bar is 1 μm. **d**, **e**, and **f** show the corresponding contours of the magnetic skyrmions.

using the optimized method, correspondingly the values are 0.27 and 0.34 using the basic method for output reconstruction. These results indicate the great potential of our physical reservoir for doing more complicated tasks with an extended reservoir size.

## Discussion

The enhancement of the recognition rate for the skyrmion state in our strain-mediated reservoir has been demonstrated in the waveform classification task, and the physical origin of the enhancement is discussed here. The output signal Hall resistivity $\rho_{xy}(E)$ is a combination of $M_z(E)$ and $\rho_{xx}(E)$, see Eq. (1). In the saturation state, the perpendicular magnetizations do not change under the variation of *E*-field, where $M_z(E)$ remains zero. So, the $\rho_{xy}(E)$ change is only attributed to the longitudinal resistivity change under *E*-field ($\rho_{xx}(E)$), which is mainly influenced by the ferroelastic strains generated by the *E*-field[49,60]. In the skyrmion state, the skyrmion number and size are controlled by the applied *E*-fields through the inverse piezoelectric effect, as shown in Fig. 5, contributing to the $M_z(E)$ in the output Hall signal, and inducing the enhancement of recognition rate. A lower density skyrmion phase is initially obtained without *E*-field applied as shown in Fig. 5a, where $\mu_0 H_a = 90$ mT. A positive *E*-field of +5 kV/cm is then applied (Fig. 5b). Evidently, a large amount of skyrmions is created, and some elongated shapes are observed. After reducing the *E*-field back to +0 kV/cm, the quantity of skyrmions is almost unchanged (note that the skyrmion winding number is unchanged due to its topological protection), while some elongated skyrmions restore to round shapes, as shown in Fig. 5c. The results indicate that strain can create skyrmions non-volatilely and deform their shapes reversibly. The binarized MFM images with extracted contours of the skymions are shown in Fig. 5d–f, where the proportion of skyrmion area ($M/M_s$) is calculated and indicated. From 0 to +5 kV/cm, the $M/M_s$ change is 23%, accompanied by skyrmion creation and deformation. When decreasing the applied voltage from +5 kV/cm back to +0 kV/cm, the $M/M_s$ change is only 6%, accompanied by a recovery of skyrmion deformation. In the waveform classification task, the input signal (*E*-field) is sweeping between +0 to +15 kV/cm repeatedly, corresponding to the phase change between Fig. 5b, c. Thus, the skyrmion deformation plays a key role for the $M_z(E)$ component in our RC system.

We further study the physical origin of the skyrmion deformation (see supplementary information S4). MOKE with rotating magnetic fields (Rot-MOKE) is conducted to evaluate the effective magnetic anisotropy fields quantitatively with the applied *E*-fields ($H_{k,eff}$). The out-of-plane magnetic anisotropy $K_{eff}$ decreases with heightened *E*-fields ($\Delta H_{k,eff} = 464$ Oe, under the *E*-field between 0 and −20 kV/cm), which is consistent with the results reported by Ba et al.[46]. The interfacial DMI constant *D* is measured by Brillouin light spectroscopy (BLS) for the Pt/Co/Gd trilayer (see supplementary information S5). Due to the out-of-plane tensile strain, a decrease of the DMI constant *D* is expected, which has previously been reported[46]. The decrease of $K_{eff}$ and the DMI constant *D* under *E*-field will enlarge and reduce the skyrmion size in the isolated skyrmion system, respectively[64]. In our experiments, the skyrmions elongate (enlarge) under applied *E*-field and restore when the *E*-field decreases to 0 (Fig. 5). Thus, we can draw the conclusion that the decrease of $K_{eff}$ plays a dominant role in the skyrmion deformation, instead of the decrease of *D* under *E*-field applied.

In the study of the MG time series prediction task, the information embedded in the small signal (peak-to-peak amplitude of a few μV) will be submerged by the system noise (which is sub μV level). Thus, we increase the output amplitude by exploiting the nonvolatile resistivity controlled by strain. In the PMN-PT(001) substrate, the ferroelectric domain structure comprises 109° and 71°/180° domains switching under applied *E*-fields in [001] direction. The in-plane strain from 109° ferroelastic switching is important for the nonvolatility in *E*-field control of the in-plane magnetization and longitudinal resistivity[48,49], which results in the memory effect in our system. After applying a negative polarized *E*-field, a loop-like response of the AHE signal $\Delta V_{xy}$ appears (see supplementary information S7), indicating that the 109° ferroelastic switching is the dominating factor in our system. In this way, the increased output signal with a magnitude of mV is sufficiently large to perform the MG prediction task against the system noise (sub μV level). Note that the anomalous Hall voltage is normally in the μV range, which barely contributes to the output signal $V_{xy}$ with a magnitude of mV. In an effort to realize the skyrmion enhancement, the magnetic tunnel junction (MTJ) can be a replacement of Hall bar device to magnify the output signal induced by skyrmion deformation, and make it comparable with the longitudinal resistivity variation with the magnitude of mV[42,49,65]. The magnitude of the tunnelling magnetoresistance (TMR) read out is estimated as following. It has been

reported experimentally that a single skyrmion would contribute a TMR signal up to 6.6 µV[65]. Considering the density of skyrmions deduced from Fig. 5c (about 60 skyrmions in the area of $5 \times 5\ \mu m^2$) and the area of our device (Hall cross $10 \times 150\ \mu m$), the number of 3600 skyrmions and the total TMR of 23.76 mV can be calculated for our RC system. The TMR read out induced by skyrmion deformation can be further estimated from the change of skyrmion area, as (26.05% −20.31%)/20.31% = 28.3% of the total skyrmion TMR change, i.e. 6.7 mV, shown in Fig. 5e, f. Hence, by employing the fusion of multiple parameters, the strain-mediated spintronic system can be considered as a powerful physical reservoir.

In addition to multi-parameter fusion, strain-mediated artificial multiferroics is advantageous for low energy consumption[43–45]. We estimate the energy dissipation in a scale-down model. The input energy (including the ferroelectric hysteresis loss) and the output read energy are taken into account (see supplementary information S8). The total power dissipation is less than 1 fJ, which is 2 orders of magnitude lower than that of other spintronic reservoirs[13,26].

Limited to the data acquisition setup, we input 10 data points per second in the experiments, which is considerably slower than previous spintronic physical reservoirs, such as spin-torque nano-oscillators[13,14]. Nevertheless, the operation frequency in such strain-mediated reservoir is potential to work at gigahertz. Due to the intrinsic ultrafast switching of ferroelectric and ferromagnetic domains[66–69], the piezoelectric and ferroelectric polarization switching time can be extremely fast down to 1 ns[70,71]. Moreover, ferrimagnetic systems exhibit ultrafast magnetization dynamics compared to ferromagnetic systems, which can reach sub-THz[68,72].

To sum up, our strain-mediated spintronic reservoir operates with all-electrical input and output voltages, exhibiting compelling ultra-low-power and multi-parameter fusion, and having the potential for ultrafast operation. To improve performance in tackling real-world problems, more effort is needed to develop the strain-mediated physical reservoir exhibiting high operation speed and computational capability with downscaled devices.

In this work, we fabricate ferrimagnetic Pt/Co/Gd multilayers on piezoelectric PMN-PT(001) substrates to host magnetic skyrmions to demonstrate a strain-mediated spintronic reservoir. E-field control of the magnetization and Hall resistivity is then investigated by MOKE and AHE, respectively, in the multiferroic heterostructure. By exploring the nonlinearity, multi-parameter fusion and short-term memory effect of this strain-mediated physical reservoir system, we successfully demonstrate a RC functionality with sequential waveform classification and Mackey-Glass time series prediction tasks. Our results show that the strain-mediated spintronic RC system enhanced by skyrmions performs better (regarding the waveform recognition rate) is mainly attributed to the strain control of magnetic anisotropy. The energy dissipation per single waveform is estimated to be on the sub-fJ scale for hybrid nanodevices. Regarding the rich characteristics of skyrmions, we believe the skyrmion-based physical reservoir has excellent potential for future research and applications. By accomplishing benchmark tasks for the skyrmion-enhanced strain-mediated RC system, our work opens a new route to low-power neuromorphic computing.

## Methods

### Sample preparation and magnetic properties characterization
Films were grown by magnetron sputtering at a base pressure below $1 \times 10^{-8}$ Torr. The sample structure was substrate/Ta(2)/Pt(3)/[Co(1.95)/Gd(1.2)/Pt(3)]$_7$ (with thicknesses given in parentheses in nanometres). The hysteresis loop measured by Polar MOKE at room temperature is shown in supplementary Fig. 2a.

L-TEM measurements with a tilting angle of 10° were conducted to observe the magnetic structures and confirm the skyrmion type in the multilayer structure at room temperature. The sample deposited on a 50-nm-thick silicon nitride membrane (CleanSiN) was used for L-TEM measurements. As shown in supplementary Fig. 2b, the magnetization reversal process, and Néel-type skyrmions in the Pt/Co/Gd multilayers are confirmed[35,46].

MFM measurements were conducted with the MFP-3D Infinity atomic force microscope, using the high-resolution and low-moment magnetic probe, SSS-MFMR (Nanosensors), with a lifted height of 30 nm.

### Electrical measurements
The electrical measurements were conducted using a Keithley 2182 A nanovoltmeter, a Keithley 6221 current source meter, a Dahua DHD60010 voltage source meter, a NF WF1948 multifunction generator, an Aigtek ATA7010 high voltage amplifier and a Keithley DMM6500 multimeter. For the waveform classification task, the input signals (0–450 V for E-field 0–15 kV/cm applied across to PMN-PT substrates) were provided by DHD60010, and the real input signal is measured by DMM6500. For the MG time series prediction task, the input signals were provided by WF1948 and ATA7010. 6221 was used to supply a 0.5 mA $I_{dc}$ current and 2182 A was employed to measure the output voltage $V_{xy}$.

### Sine and square waveform classification
For the sequence waveform classification task in Fig. 3, the input is a random waveform sequence comprising 400 sine and square waveforms with the same period around 9 s. We use the five-fold cross-validation technique to estimate the performance of the task. The training goal is to determine a set of weights $w_j$, where $j$ is the index for $n$ sampled values. These weights are used to multiply the $n$ sampled output weights to give an output value $y$, which should ideally equal to one for square waves and zero for sine waves. Note that $w_j$ is different for recognizing the current and past input signals. For the training process, we use a technique called the linear Moore-Penrose pseudo-inverse operator to extract the eigenvalues from singular matrices. Specifically, considering the matrix target $Y$ containing all the targets and $S$ containing all the output signals for training, the weight matrix containing all the optimal weights of $w_j$ is given by $W = YS^{\dagger}$, where the dagger symbol denotes the pseudo-inverse operator. Note that the weights training is performed separately for each of the different reservoirs (skyrmion and saturation states).

### Mackey-Glass time series prediction
The dataset for the prediction task is prepared in the following way. First, a Mackey-Glass time series is obtained by solving Eq. (2). For the experiment, we obtain 2500 data points in total. After removing the results of the initial 600 data points (for the initialization of the reservoir), the first 1200+$i$ data points are used for training, and the rest 700−$i$ are for testing, where $i$ is the prediction horizontal step. The first stage of the masking procedure is a matrix multiplication $W_{in} \cdot M_o$, where $W_{in} \in \mathbf{R}^{Nr \times 1}$ is the mask matrix with data values drawn from a standard normal distribution and $M_o \in \mathbf{R}^{1 \times L}$ is the original input data. Here $L = 1200+i$ is the length of the original MG data points and $N_r$ is the reservoir size. We adopt $N_r = 50$ in the experiment. As a consequence of the masking, we obtain the data matrix $M_e = W_{in} \cdot M_o \in \mathbf{R}^{Nr \times L}$. Then $M_e$ is column-wise flattened into a vector $e \in \mathbf{R}^{Nr \cdot L}$ and then fed into the reservoir of skyrmion systems.

Each of the value from $e$, multiplied and translated as the E-field (ranging from 0 to 5 kV/cm as shown in Fig. 4a), is provided as pre-processed input into the reservoir with a time interval $t_{step} = 0.1$ s. In the following, the reservoir dynamics is recorded for every $t_{step}$ to form a vector of $M_y \in \mathbf{R}^{Nr \cdot L}$, which is then unflattened into a response matrix $M_x \in \mathbf{R}^{Nr \times L}$ for output reconstruction. We use the matrix $A \in \mathbf{R}^{Nr \times Tr}$ consisting the first $T_r = L-i$ columns of $M_x$ for training the read out matrix. The teaching matrix $B \in \mathbf{R}^{1 \times Tr}$ consisting the last $T_r$ of the original signal $M_o$ is the time series to be predicted. A read-out matrix $W_{out}$ is

therefore constructed through the method of ridge regression,

$$W_{\text{out}} = (A \cdot A^{\text{T}} + \mu I)^{-1} (A \cdot B^{\text{T}})$$

where $\mu = 10^{-4}$ is used as regularization parameter.

Instead of using only the reservoir's output from the current step, we concatenate the response matrix $A$ at each column with output states from previous steps to form a new matrix $A^* \in \mathbf{R}^{(Nr \cdot (1+np)) \times Tr}$, where $n_p$ is the number of previous steps' states.

## Data availability
The data that support the findings of this study are available within the paper and the Supplementary Information. Additional data related to this study are available from the corresponding authors upon reasonable request. Source data are provided with this paper.

## Code availability
The custom code used in this study are available from the corresponding authors upon reasonable request.

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

## Acknowledgements

This work was mainly supported by the National Natural Science Foundation of China (NSFC) Grants No. 52061135105 and 12074025 received by N.L. W.K. acknowledges the supports from the Beijing MSTC Nova Program (Z211100002121014 and Z201100006820042), and the NSFC Grant No. 62274008, and Beijing Natural Science Foundation (L223004). D.W. acknowledges the support from the Chinese Academy of Sciences (CAS) Project for Young Scientists in Basic Research (Grant No. YSBR-030). C.H.B. acknowledges fundings by the German Research Foundation (DFG) via project BA 2181/21-1 and the excellence cluster MCQST under Germany's Excellence Strategy EXC-2111 (Project No. 390814868). J.Z. acknowledges the support from the Research Equipment Development Project of CAS (YJKYYQ20210001). J.H.Z. acknowledges the support from the NSFC Grant No. 11834013. M.F. acknowledges the supports from the NSFC Grants No. 51772126 and 52171210.

## Author contributions

N.L. conceived the project and designed the research with the help of W.K. Y.S. and T.L. prepared the sample with help from Z.Z., D.W., and J.H.Z. Y.S. performed MOKE, and AHE measurements with help from T.L. and C.C. Y.S. performed MFM measurements with help from L.H. and M.F. T.L., C.C., and L.Y. performed SQUID-VSM measurements. T.L., E.D., S.P., and J.Z. performed the BLS measurement. L.L. and D.P. conducted the cross-sectional TEM measurement. A.U. and C.H.B. conducted the L-TEM measurement. Waveform classification and Mackey-Glass time-series prediction tasks were realized by Y.S., X.C., and W.K. L.Z., Z.Y., D.W., A.F., and W.Z. were involved in the discussion. All the authors contributed to analysing the results and writing the manuscript.

## Competing interests

The authors declare no competing interests.
