## [Peer Review File · Nature Communications]

Experimental Demonstration of a Skyrmion-Enhanced Strain-mediated Physical Reservoir Computing SystemREVIEWER COMMENTS

Reviewer #1 (Remarks to the Author):

The paper by Sun et al. presents a skyrmion-enhanced straintronic physical reservoir in a multiferroic heterostructure of Pt/Co/Gd multilayers on (100)- oriented $0.7\text{PbMg}_{1/3}\text{Nb}_{2/3}\text{O}_3$ - 0.3PbTiO_3 (PMN-PT). It is an interesting idea and worthwhile effort to explore new scenarios of applications for such strain-mediated multiferroic heterostructures, but I would not be able to recommend its publication in Nature communications for the following reasons.

First, the paper reads more like "here is the Hall bar measurement data, and it can mimic functions for reservoir computing, and that is exciting". Most of content of the paper was spent on describing the experimental observation. I did not see a compelling narrative on why strain-mediated artificial multiferroics is better than other competing technologies for Reservoir computing, and there seem to be no rigorous benchmarking experiments against other competing technologies. Without such benchmarking results, I am not convinced that the paper can stimulate wide interests in the community of magnetic and ferroelectric materials or the neuromorphic computing. Furthermore, it seems that the idea of using Skyrmions for reservoir computing is already there, and the only advancement in this work is to demonstrate the electric-field control for claiming lower-power, but that is a somewhat natural expectation. In addition, the energy dissipation of such system should mainly come from the ferroelectric hysteresis loss, and thus the method the author used in Supplemental for estimating the energy dissipation is incorrect.

Second, the paper offers little to no explanation or insights into the underlying physical principles. Is it mainly based on strain control of interfacial DMI or anisotropy? What is the role of ferroelectric domain structure? In Fig. 3b (which seems to be one of the key results), Why the labyrinth domain and the saturated state respond similarly to the input signal? Why the Skyrmion state respond to the input signal more strongly?

Third, it seems that the three physical reservoirs the authors mentioned (labyrinth domain, saturated state, and skyrmions) cannot coexist, and that they cannot be switched from one state to another by electric field. Rather, they need to be changed by changing the bias magnetic field. Would that influence the Reservoir computing application?

Then there are other problems like why a ferrimagnetic multilayer Pt/Co/Gd was used rather than ferromagnetic multilayer. For a serious device design paper that warrants publication in Nature communications, materials design (what materials, how materials parameters affect the device performances) and device design principles should all be clearly articulated.

Overall, It reads to me more like a pure application-driven work, and should be more suitable for consideration for publication in applied physics journal.

Reviewer #2 (Remarks to the Author):

The authors demonstrate reservoir computing (RC) using AHE response to an input voltage applied to a PMN-PT (100) substrate on which Pt/Co/Gd multilayers are deposited. They explore magnetization in the heterostructure film is in saturated, skyrmion and labyrinth states and explain the reason behind the skyrmion state being most effective for RC, which is its being most responsive to the E-field. (This claim for the AHE responsiveness to E-field is corroborated by MFM images under applied E-field in the skyrmion state in the supplement).

They further show that with appropriate training of weights of the output layer of their RC system they can classify waveforms and predict Mackey-Glass time series. Overall, I think

this work is an important contribution to the field of physical reservoir computing using spintronics/nanoscale magnetism. However, before it is suitable for publication several important issues need to be addressed.

Important Issues:

1. Strain generation by applying a electric field: Figure 1 shows that voltage is applied via a back-gate while a top electrode in the corner of PMN-PT substrate is grounded. Considering PMN-PT is an insulator how can there be field lines perpendicular to the PMN-PT substrate unless the whole top surface is grounded. (If a complete top electrode is applied and grounded it would not allow the transport/AHE measurements).

2. One possibility is that the magnetic state is also sensitive to small magnetic fields (in addition to strain). Did the authors ensure that is the process of applying a time varying voltage to the PMN-PT substrate, they did not also produce a small magnetic field and the RC was a result of this and not the E-field.

Other Issues

1. I do not see how the Neural ODE adds any value to the key message of the paper, which is demonstrating that the AHE output due to the voltage induced strain can implement RC effectively in the skyrmion state. Whether the AHE (magnetization) response can be predicted/simulated by Neural ODE/micromagnetic does not prove/disprove the efficacy of this system for RC.

2. With E field perpendicular to the substrate and along the poling direction, the PMN-PT (100) substrate would be isotropic in-plane. Hence, the effect of strain is modulating the PMA. Could the authors clarify this?

Jayasimha Atulasimha
Virginia Commonwealth University.
(signed as I am comfortable doing so for transparent review)

Reviewer #3 (Remarks to the Author):

The primary contributions of this manuscript are:

- (1) the first experimental demonstration of a reservoir computer based on nanomagnetic textures with electrical inputs; and
- (2) the first simulation of a skyrmion reservoir computer performing a time-series forecasting task.

This manuscript also includes:

- (3) analysis of the experimental and simulation results, including an estimate of energy dissipation.

My review of each of these contributions follows:

(1) The experiment convincingly demonstrates that the skyrmion reservoir computer indeed functions as a reservoir. However, this reservoir seems to function considerably slower and provide reduced computational capabilities relative to alternative reservoirs. To enable evaluation of this reservoir's computational capabilities, the authors should report its parity check capacity and short-term memory content; this calculation may be performed through further analysis of the experimental data that has already been presented.

(2) The authors have used the neural ODE approach to demonstrate that their system can

implement the Mackey-Glass time series prediction task; however, while this neural ODE approach can give design guidance and provide a rough estimate of how it will work IF it works, it cannot be considered a reliable and authoritative prediction of system functionality in a manner similar to micromagnetic simulations. Therefore, the authors cannot rely solely on this neural ODE simulation in order to prove that this reservoir computer can perform a time-series forecasting task. As their experimental setup appears suitable for demonstration of the Mackey-Glass task, why didn't the authors perform this task experimentally?

(3) The authors estimate that the energy dissipation for the waveform classification task is less than 1 fJ per waveform. However, this estimate does not include the energy required to read the output. In many physical reservoir proposals, the output read energy is far larger than the input energy; I expect this to be the case for the authors' proposed system as well. It is therefore critical that the authors consider the output read energy, which will likely dominate the total energy dissipation estimate.

Reviewer #4 (Remarks to the Author):

This paper reports the strain enhancement due to the formation of magnetic skyrmions in PMN-PT. They performed benchmarking using Mackey-Glass series. This can be useful for reservoir computation with low-power and high-speed operation as the authors pointed out. This paper can be published in Nat. Commun. after considering the following issues.

(1) The authors used the term of "straintronics" throughout their manuscript. Unfortunately this term is not commonly used in the community and is needed some clear definition in the introduction.

(2) In Fig. 1, the meaning of the blue "Straintronic reservoir" diagram is not clear. The authors should also include the dimensions of their device in Fig. 1. They may wish to include an optical micrograph or scanning electron micrograph of their device in their supplemental information. They should also explain the width and amplitude of the input signals in the caption. They should provide a clear MFM image with a scale bar showing the magnetic force in their supplement.

(3) The authors should add the details of the quality of their devices. Especially they should add a cross-sectional transmission electron micrograph to confirm the quality of the interfaces, which are crucial to determine the skyrmion formation.

(4) In Fig. 2a, the authors showed two MFM images as insets. Were they taken at the same position? It would be better to extract the density of the skyrmions from these images to support their discussion as well as the size distributions of the skyrmions under different magnetic fields.

(5) The authors should comment on the origin of the minor hysteretic behaviour, i.e., two peaks near the zero electric field under the magnetic field application of 90 mT, observed in Fig. 2b.

(6) It would be helpful for the readers to include a schematic diagram to show the lattice stretch with % estimated under representative electric fields. Please also discuss the reproducibility and any fatigue to be introduced after a certain operation cycle.

(7) The authors should quantify the statement on line 151, "... roughly ..."

(8) Would the authors be able to change the numbers of skyrmions to be generated monotonically?

(9) On line 178, the authors should clarify the meaning of "the short-term memory". What is the time scale they envision?

(10) It might not be so clear for the readers how the authors would use the skyrmions for reservoir computation as schematically shown in Fig. 3a. How advantageous would their device be as compared with the other concepts for the reservoir computation, such as in terms of their operation range?

(11) On line 197, please quantify "a small amount of sample data".

(12) In the conclusion, the authors commented the operation frequency can reach GHz. Please include any supporting data.

(13) Some skyrmions in Figs. S2f and g seem to be deformed. Have the authors confirmed if these are really skyrmions (not maze domains and/or magnetic bubbles)? Please also add comments if they can treat such deformed skyrmions in the same way as the circular ones for their reservoir computation.

Manuscript ID: NCOMMS-22-35045

Manuscript title: Experimental Demonstration of a Skyrmion-Enhanced Strain-mediated Physical Reservoir Computing System

Authors: Yiming Sun, Tao Lin, Na Lei, Xing Chen, Wang Kang, Zhiyuan Zhao, Dahai Wei, Chao Chen, Simin Pang, Linglong Hu, Liu Yang, Enxuan Dong, Li Zhao, Lei Liu, Zhe Yuan, Aladin Ullrich, Christian H Back, Jun Zhang, Dong Pan, Jianhua Zhao, Ming Feng, Albert Fert, and Weisheng Zhao

Response Letter

We gratefully thank the four referees for their precious time and insightful comments. We found all their comments constructive and helpful. In particular, we appreciate Referee #1 for the comments “*It is an interesting idea and worthwhile effort to explore new scenarios of applications for such strain-mediated multiferroic heterostructures*”, and we appreciate Referee #2 for the comments “*this work is an important contribution to the field of physical reservoir computing using spintronics/nanoscale magnetism*”, and we appreciate Referee #3 for the comments “*the first experimental demonstration of a reservoir computer based on nanomagnetic textures with electrical inputs*”, and we appreciate Referee #4 for the comments “*This can be useful for reservoir computation with low-power and high-speed operation as the authors pointed out. This paper can be published in Nat. Commun. after considering the following issues*”. We have taken all review comments into account and have carefully revised the paper, including a deeper physical discussion, more comprehensive energy estimation and the benchmarking Mackey-Glass time series prediction experiment. The manuscript has been significantly improved after the revision, and we hope it now satisfies the publication requirements of *Nature Communications*. Below we provide our point-to-point responses to all comments. For the referees’ convenience, we present comments in blue color, followed by our responses in black. The major changes in the revised manuscript are highlighted in red.

Referee's General Remarks:

The paper by Sun et al. presents a skyrmion-enhanced straintronic physical reservoir in a multiferroic heterostructure of Pt/Co/Gd multilayers on (100)- oriented 0.7PbMg1/3Nb2/3O3-0.3PbTiO3 (PMN-PT). It is an interesting idea and worthwhile effort to explore new scenarios of applications for such strain-mediated multiferroic heterostructures, but I would not be able to recommend its publication in Nature communications for the following reasons.

Overall, It reads to me more like a pure application-driven work, and should be more suitable for consideration for publication in applied physics journal.

Authors' Reply:

We sincerely appreciate the referee's comment on our paper as "*an interesting idea and worthwhile effort to explore new scenarios of applications for such strain-mediated multiferroic heterostructures*". Based on the referee's suggestions, we improved our work with more emphasis on the analysis of the physical mechanism and a deeper discussion. We believe that it is now clearer to understand the underlying phenomena of the strain-mediated physical RC device. The characters of the nonlinearity and the short-term memory effects for RC are discussed, which are attributed from the ferroelectric domains switching ($71^\circ/180^\circ$ and 109°). Further, the skyrmion enhancement of the RC performance is attributed to its spin texture deformation and recovery induced by strains, which is mainly due to magnetic anisotropy changes through magnetoelastic effects. We have added a *discussion* section in the revised manuscript for the physical origins of the strain-mediated physical RC.

A recurrent neural network is a competent network architecture for complex tasks, which has great potential in neuromorphic-based applications. In this work, we used strain-mediated voltage-control of the magnetization and resistivity change to realize physical reservoir computing (RC). We achieved the waveform classification task and the benchmarking Mackey-Glass time series prediction task. Thus, our work has broadened the application prospects for strain-mediated spintronic devices in the field of artificial intelligent computing. To date, this is the first experimental demonstration of spintronic RC with electronic voltages as both input and output signals.

Additionally, the strain-mediated RC is inherently capable for multi-parameter fusion and potentially for ultra-low-power consumption (the energy dissipation of our RC is 2 orders of magnitude lower than the reported ones [*Nature* **547**, 428 (2017)]). This work performs the RC system in the view of not only applications but also the insights of physical principles, and thus we believe that the current version satisfies the requirements for publication in *Nature Communications*.

Referee's Comment #1:

First, the paper reads more like "here is the Hall bar measurement data, and it can mimic functions for reservoir computing, and that is exciting". Most of content of the paper was spent on describing the experimental observation. I did not see a compelling narrative on why strain-mediated artificial multiferroics is better than other competing technologies for Reservoir

computing, and there seem to be no rigorous benchmarking experiments against other competing technologies. Without such benchmarking results, I am not convinced that the paper can stimulate wide interests in the community of magnetic and ferroelectric materials or the neuromorphic computing. Furthermore, it seems that the idea of using Skyrmions for reservoir computing is already there, and the only advancement in this work is to demonstrate the electric-field control for claiming lower-power, but that is a somewhat natural expectation.

Authors' Reply:

We sincerely thank the referee for his/her instructive comments, which make our work more comprehensive. We added a *discussion* section in the revised manuscript to give insight into the physics of the system described in our manuscript. To strengthen the paper, we additionally performed the Mackey-Glass (MG) time series prediction task experimentally and attain rigorous results.

As to the advantages using “*strain-mediated artificial multiferroics*” for RC, we give the reasons as following:

Firstly, the strain-mediated magnetic system fits the nonlinearity naturally (the relation between strain and applied electric field), which is a key precondition for physical RC system, as shown in Fig. 2c.

Secondly, strain-mediated artificial multiferroics for controlling magnetization switching are advantageous for low energy consumption [*Nat. Commun.* **4**, 1378 (2013); *Nat. Mater.* **16**, 712 (2017)]. As shown in Fig. R1, strain-mediated spintronic devices dissipate considerably low switching energy compared to other driving methods.

Fig. R1 Experimental demonstration and theoretical prediction of switching energy for various mechanisms. The strain-mediated spintronic devices are more energy efficient than most spintronic devices. [Wang, J.P., et al., *Proceedings of the 54th Annual Design Automation Conference*, 1-6 (2017)]

In the hardware-based magnetic reservoirs, e.g., in ref. [*Appl. Phys. Lett.* **118**, 202402 (2021)], the estimated operation energy is 24 fJ per input. In our strain-mediated spintronic RC system, the energy dissipated for each input waveform is estimated to be sub-Femto-Joule in nano scale devices, which is two orders of magnitude lower than other spintronic reservoirs, see Table 1 shown below.

Table 1 | Estimated power consumption for implementing the reservoir using different systems.

Reservoir type	Energy/Input	Reference
CPU-based	54.8 μ J	Nat. Electron. 2 , 480 (2019).
FPGA-based	143 nJ	
Memristor-based	3.0 nJ	
Spin-torque nano-oscillator/MTJ-based	\sim 100 fJ (1 μ W)	Nature 547 , 428 (2017).
Strain-mediated voltage-controlled super-paramagnet-based	24 fJ	Appl. Phys. Lett. 118 , 202402 (2021)
Strain-mediated Skyrmion-based	0.85 fJ	★This work

Thirdly, strain is a universal way to control various characteristics in physical systems as shown in Fig. R2 [*Phys.-Usp.* **61** 1175 (2018)], which have been widely studied, including magnetization, resistance, Dzyaloshinskii–Moriya interaction (DMI), phase transitions, luminescence, etc. Hence, multiferroic heterostructures inherently capable for multi-parameter fusion are promising to be a powerful physical reservoir for different tasks. In our work, we proposed a strain-mediated Hall bar device using the electric-field (E -field) as inputs and the anomalous Hall effect (AHE) response as outputs, where the longitudinal resistivity and magnetization from the skyrmion deformation are tuned by E -fields simultaneously. We experimentally realize a skyrmion-enhanced strain-mediated physical reservoir by combining the magnetization and resistivity changes. Furthermore, we demonstrate its functionality via a sequential waveform classification task and a Mackey–Glass time series prediction task. We have added the description in the introduction part of the revised manuscript “(lines 71-75) *Moreover, strain is a universal way to control various characteristics, which have been widely studied, including magnetization [43-48], resistance [49,50], Dzyaloshinskii–Moriya interaction (DMI) [51], phase transitions [52], luminescence [53], etc.*”.

Fig. R2 Two branches of straintronics. Strain-mediated manipulation of the electronic structure (left part), and main magnetic straintronic devices (right part). [*Phys.-Usp.* **61** 1175 (2018)]

Next, regarding “*the idea of using Skyrmions for reservoir computing*”:

Magnetic skyrmions exhibiting small size (sub-10 nm), high energy efficiency, and especially topological stability show exclusive advantages for implementing a reservoir regarding nonlinear response, memory of past manipulations and complex interaction between multiple skyrmions. Thus, several simulation works concerning skyrmion-based reservoirs have been proposed since 2018 [*Phys. Rev. Appl.* **9**, 014034 (2018); *AIP Adv.* **8**, 055602 (2018); *Phys. Rev. Appl.* **14**, 054020 (2020); Preprint at <https://arxiv.org/abs/2108.01512> (2021); *Neuromorph. Comput. Eng.* **2**, 044011 (2022)]. However, to realize the demonstration of skyrmion-based RC experimentally is far from easy.

Only recently, three experimental demonstrations are published or preprinted on arXiv. A Brownian RC was demonstrated in a geometrically confined skyrmion reservoir based on its nonlinear and stochastic dynamics [*Nat. Commun.* **13**, 6982 (2022)], pattern recognition was realized by skyrmion dynamics induced by magnetic-fields [*Sci. Adv.* **8**, eabq5652 (2022)], and a task-adaptive approach has been researched in a chiral magnet that hosts different magnetic phases including skyrmion, helical and conical [Preprint at <https://arxiv.org/abs/2209.06962> (2022)]. However, the skyrmion location determined by optical imaging as output is used in Brownian RC, magnetic fields as inputs are utilized both in the pattern recognition work and the task-adaptive physical RC. One should stress that nonelectrical signals as both inputs and outputs are far from further applications. Instead, our strain-mediated Hall bar device using E -field as inputs and AHE response as outputs are all electrical voltage signals, which is a step forward for physical RC applications.

The RC manipulated by strain has been proposed by Welbourne, A., et al. [*Appl. Phys. Lett.* **118**, 202402 (2021)], demonstrating the possibility to perform benchmark machine learning tasks with low energy costs. However, the experimental implementation is still lacking. Here, we present the first demonstration of strain-mediated spintronic RC. It can reduce the power consumption which is a core issue for practical application, and we studied the physical mechanisms for its voltage response.

We now have succeeded to perform “*rigorous benchmarking experiments*” (details below).

MG time series prediction as one of the most representative benchmark tasks for RC, has been demonstrated experimentally and added in the revised manuscript. The pre-processed input signal and the output V_{xy} is shown in Fig. R3a (the grey and the red data sets, respectively). Utilizing the nonlinearity and short-term memory of the strain-mediated system, the predicting results in terms of the prediction error is evaluated by a normalized root mean square error (NRMSE), which can reach 0.2 for a 20-step prediction, as shown in Fig. R3b (the same figure in the revised manuscript as Fig. 4b).

Figures R3c to R3f show the predicting results of MG time series for the next value ($i = 1$), and the value happening 20 steps later ($i = 20$), using the basic method and the optimized algorithm (extending the reservoir size by using previous steps’ output states) for output reconstruction. Each horizontal prediction step i corresponds to a different prediction task. In the figures, the blue curve is the ground truth, the black and the red curves are prediction results from basic and optimized methods, respectively. The prediction error apparently increases in Fig. R3d compared to R3c, and also the same trend shown in Fig. R3f compared to R3e. Additionally, the predicted results using the optimized algorithm shown in Figs. R3e and R3f demonstrate better accuracy than those using

the basic method shown in Figs. R3c and R3d, respectively. To be specific, the NRMSE values are 0.16 and 0.2 for predictions of $i = 1$ and $i = 20$ using the optimized method, correspondingly the values are 0.27 and 0.34 using the basic method for output reconstruction. The results indicate the great potential for our physical reservoir with extended reservoir size.

Fig. R3 Mackey-Glass time series prediction task demonstrated by the strain-mediated spintronic RC system. **a**, Input pre-processed signal (grey data set) and the corresponding output signal (red data set) of the strain-mediated reservoir. **b**, NRMSE as a function of horizontal prediction step i , shown in log scale, for the testing set by using the basic method (black curve) and the optimized algorithm (red curve) for output reconstruction. **c** and **d** are the selected predicting results (black) for horizontal step $i = 1$ (for short-term prediction) and 20 (for long-term prediction), respectively, predicted by the strain-mediated RC system using basic method for output reconstruction, in comparison with the ground truth (blue) of MG time series. **e** and **f** are the selected predicting (red) results for horizontal step $i = 1$ and 20, respectively, predicted by the strain-mediated RC system using optimized algorithm for output reconstruction, in comparison with the ground truth (blue) of MG time series.

We have revised the manuscript “**Strain-mediated RC System for Mackey-Glass time series prediction task**” part (lines 213-275).

At last, our work is the only experimental demonstration with all the electrical input and output signals based on spintronic devices up to now, which benefits for real application of the physical RC presently.

In sum, we are confident that our study will promote the application of strain-mediated spintronic devices in neuromorphic computing, and it is also of general interest due to multi-parameter fusion and ultra-low power consumption.

Referee's Comment #2:

In addition, the energy dissipation of such system should mainly come from the ferroelectric hysteresis loss, and thus the method the author used in Supplemental for estimating the energy dissipation is incorrect.

Authors' Reply:

Thank you very much for pointing out this problem. The ferroelectric hysteresis loss is determined by integrating the area of the P - E hysteresis loop using Greene's theorem. In our experimental process, the unipolar sweeping of the E -fields was performed. For (001)-oriented PMN-30%PT, which is the same substrate as we used in the RC device, the unipolar P - E response is shown in Fig. R4a. Since barely hysteretic P - E responses driven by the unipolar electric field, the total hysteresis loss is estimated to be less than 2000 J/m^3 per cycle, as shown in Fig. R4b [*J. Am. Ceram. Soc.* **89**, 775 (2006)]. Considering a 100 nm diameter nanomagnet on a 100 nm thick PMN-PT (as discussed in the supplementary information S8), the estimated energy dissipation from ferroelectric hysteresis loss will be less than 6.28 aJ, which is 2 orders of magnitude smaller than the operating energy cost estimated in supplementary information S8 (850 aJ). Therefore, we can ignore the ferroelectric hysteresis loss when estimating the energy dissipation. We have revised supplementary information S8 to make the energy estimation more accurate.

Fig. R4 Unipolar P - E response and hysteresis loss for (001)-oriented crystal of PMN-PT under different axial stresses between 0 and $6 \times 10^7 \text{ N/m}^2$. a, P - E response. b, hysteric loss. [*J. Am. Ceram. Soc.* **89, 775 (2006)]**

Referee's Comment #3:

Second, the paper offers little to no explanation or insights into the underlying physical principles. Is it mainly based on strain control of interfacial DMI or anisotropy?

Authors' Reply:

We would like to thank the referee for the valuable comments. In the revised manuscript, we have added a *discussion* part which includes but is not limited to the underlying physical principles.

We attribute the underlying physical mechanisms of the skyrmion deformation to the decrease of magnetic anisotropy K_{eff} instead of DMI constant D under applied E -field (controlled by strain), as discussed below.

MOKE with rotating magnetic fields (Rot-MOKE) was conducted to evaluate the effective magnetic anisotropy fields quantitatively with the applied E -fields. For Rot-MOKE experiments, we started rotating from the polar-MOKE configuration as illustrated in Fig. R5a. Figure R5b shows the experimental (data points) and fitting (curves) results of the torque $l(\theta)$ as a function of the magnetization equilibrium angle θ under different applied E -fields. The effective magnetic anisotropy fields ($H_{k,\text{eff}}$) are -267 , -504 and -731 Oe for 0 , -10 and -20 kV/cm, respectively. A negative $H_{k,\text{eff}}$ is obtained, which means that there is an in-plane magnetic anisotropy in Pt/Co/Gd multilayers. The in-plane magnetic anisotropy presumably originates from the strong dipolar interaction due to the multi-repetitions, since the trilayer of Pt/Co/Gd shows clear out-of-plane easy axis loop in Fig. R5c. The change of magnetic anisotropy fields $\Delta H_k(E) = H_{k,\text{eff}}(E) - H_{k,\text{eff}}(E_0)$ is plotted as a function of the sweeping E -fields, shown in Fig. R5d. A butterfly shape (typical strain- E relationship of PMN-PT substrates) with the same switching fields (± 1.67 kV/cm) is shown in Fig. 2b, meaning that the anisotropy change originates from the strain generated by E -fields.

Here, the effective magnetic anisotropy is $K_{\text{eff}} = K_u - \frac{\mu_0 M_s^2}{2} = \frac{\mu_0 M_s H_{k,\text{eff}}}{2}$, where K_u is the uniaxial magnetic anisotropy, μ_0 is vacuum permeability and M_s is the saturation magnetization. Utilizing the strain-mediated magnetoelectric coupling in the piezoelectric/ferrimagnetic heterostructure, the simplified strain-induced anisotropy is written as $K_e = -\frac{3}{2}\lambda\sigma$, $\sigma = \varepsilon E_f / (1 - \nu^2)$, where λ is the magnetostriction coefficient, ε is the strain, E_f is the Young's modulus, and ν is the Poisson ratio [*Phys. Rev. B* **84**, 012404 (2011)]. For Pt/Co/Gd multilayers, we took $E_f = 168$, 209 , and 55 GPa for Pt, Co and Gd to get the effective Young's modulus, $\nu = 0.3$ for metals and $\lambda = -1.7 \times 10^{-4}$ for GdCo₂ alloy [*Phys. Stat. Sol. (a)* **34**, 383 (1976); <https://periodictable.com/Properties/A/YoungModulus.html>]. For PMN-PT (100) single crystal, $\varepsilon = 0.2\%$ at ± 20 kV/cm was taken [*Appl. Phys. Lett.* **78**, 2551 (2001)]. The strain-induced anisotropy change in Pt/Co/Gd multilayer can be estimated: $K_e = 8.5 \times 10^4$ J/m³. The equivalent anisotropy field change is $\Delta H_e = \frac{2K_e}{\mu_0 M_s} = 1416.7$ Oe, where the measured $M_s = 1.2 \times 10^6$ A/m. The measured $\Delta H_{k,\text{eff}} = 464$ Oe (between 0 and -20 kV/cm) is lower than the estimated value, which is presumably due to the transferred strain loss in the multilayers compared to the PMN-PT substrate. Additionally, for the positive piezoelectric coefficients d_{33} in PMN-PT, a tensile strain in out-of-plane direction is generated when the E -field is higher than the ferroelectric switching field, resulting in an in-plane compressive strain. Considering the strain transferred from PMN-PT to the multilayers on top, together with the negative λ in CoGd, a decrease of the out-of-plane magnetic anisotropy can be concluded. Namely, the in-plane magnetic anisotropy increases with E -fields, which is consistent with the measured results, as the negative ΔH_k shown in Fig. R5d.

Fig. R5 Evaluation of magnetic anisotropy changes with E -fields. **a**, Sketch of Rot-MOKE measurements. **b**, The torque $l(\theta)$ versus θ with different E -fields, the solid curves are fitting results. **c**, Out-of-plane hysteresis loop for the Pt/Co/Gd trilayer sample, where the thickness of the Co layer is 1.95 nm. **d**, Fitting results of the magnetic anisotropy field change ΔH_k as a function of the E -fields. The arrows indicate the E -field sweeping direction.

The out-of-plane magnetic anisotropy decreases with heightened E -fields, which is consistent with the results reported by Ba, Y., et al. [*Nat. Commun.* **12**, 322 (2021)] as shown in Fig. R6a. Since our system used the same substrate and Pt/Co based multilayer structure, the strain-induced anisotropy change in our system is measured to be $9 \times 10^3 \text{ J/m}^3$ at -4 kV/cm (in-plane compressive strain), which is also comparable to the result shown in Fig. R6a ($2.5 \times 10^4 \text{ J/m}^3$ for [Pt/Co/Ta]₅ at -4 kV/cm). We have added the results of anisotropy change versus strain in supplementary information S4.

Fig. R6 Strain-mediated voltage-control of interfacial DMI and magnetic anisotropy. **a**, K_{eff} versus electric-field curve. **b**, Interfacial DMI value D versus electric-field curve. [*Nat. Commun.* **12**, 322 (2021)]

As shown in Fig. R6, the interfacial DMI also changes with E -fields and shows similar behavior as the magnetic anisotropy (K_{eff}). In this work, [Pt/Co/Ta]₅ is studied, the interfacial DMI stems mainly from the Pt/Co interface since the Ta/Co interface generates very weak interfacial DMI [*Nat. Mater.* **15**, 501 (2016)]. Referred to our studied system of [Pt/Co/Gd]₇, the interfacial DMI is also mainly from the Pt/Co interface since the orbital moment is zero in Gd, whose spin-orbit coupling is negligible [*J. Phys. Condens. Matter.* **3**, 5131 (1991)]. In Fig. R6b, a reduction of ~24% under an applied E -field (-4 kV/cm) can be seen, which stems from the weakening of the Pt-Co hybridization at the interface induced by the out-of-plane tensile strain. Thus, a reduced D in our [Pt/Co/Gd]₇ films can be expected with the out-of-plane tensile strain. If we assume that the reduction rate is comparable to the one shown in Fig. R6b, the change interval would be -0.22~-0.29 mJ/m² in our system, taking the measured DMI constant D of -0.29 mJ/m² in Pt/Co/Gd trilayers (supplementary information S5). Note that D in the multi-periods of Pt/Co/Gd should be the same as Pt/Co/Gd trilayer, because each magnetic layer has the same adjacent layers, which is supported by the result in Ba, Y.'s work [Fig. S8b in *Nat. Commun.* **12**, 322 (2021)].

Based on the MFM images in the manuscript, our system hosts isolated skyrmions instead of a skyrmion lattice. For the isolated skyrmions, the skyrmion size is related to K_{eff} and D : $R = \pi D \sqrt{\frac{A}{16AK^2 - \pi^2 D^2 K}}$ [*Commun. Phys.* **1**, 31 (2018)]. The decrease of K and increase of D will enlarge the skyrmion size. In our experiments, the skyrmions elongate (enlarge) under applied E -field and restore when the E -field decreases to 0 (see Fig. R7). Thus, we can draw the conclusion that the decrease of K_{eff} plays a dominant role in the skyrmion deformation, instead of the decrease of D under applied E -field.

Fig. R7 MFM images at different E -fields. [Pt/Co/Gd]₇ in the skyrmion state. The MFM image size is $5 \times 5 \mu\text{m}^2$.

Referee's Comment #4:

What is the role of ferroelectric domain structure?

Authors' Reply:

We gratefully thank the referee for this valuable comment.

The ferroelectric domain structure can induce polarization charges at the PMN-PT surface, and also generates strain in PMN-PT. The coupling between ferroelectric and ferromagnetic layers are mainly from the charges and strains.

Due to the polarization charges located at the PMN-PT surface, the interfacial coupling between ferroelectric domains and magnetic domains has been studied [*Adv. Mater.* **23**, 3187 (2011)];

Phys. Rev. Lett. **107**, 217202 (2011)]. The magnetic stripe domains of the CoFe film (FM) are induced by the ferroelectric domain structure (FE) as shown in Fig. R8, where a CoFe film is grown directly on the ferroelectric substrate BaTiO₃ without buffer layer. However, the charges at the FE/FM interface have only a few angstroms screening length [*Phys. Rev. B* **75**, 054408 (2007)]. Our multilayers film structure includes 2 nm Ta and 3 nm Pt as buffer layers, which is far beyond the charge-screening length. Therefore, a direct coupling between ferroelectric and magnetic domains is negligible in our PMN-PT/ferrimagnetic multilayers heterostructure.

Fig. R8 Magnetic hysteresis curve and polarization microscopy images of the ferroelectric domain structure (FE) and magnetic stripe domain in the CoFe film (FM) during several stages of the magnetization reversal process. [*Adv. Mater.* **23**, 3187 (2011)]

In the PMN-PT(001) substrate, the ferroelectric domain structure comprises 109° and 71°/180° domains switching under E -fields applied along [001] directions. As shown in Fig. R9a [*Phys. Rev. Lett.* **108**, 137203 (2012)], when the poling directions change from positive to negative, there are two kinds of ferroelectric domains switching (71°/180° and 109°). In 71°/180° domain switching, the in-plane distortions along the [110] and [-110] directions keep unchanged, with elongation along [110] and compression along [-110]. While for 109° domain switching, the in-plane distortions along the [110] and [-110] directions swap, i.e. from elongate distortion to compression along [110], and reversely along [-110].

The in-plane strain from 109° domain switching induces a non-volatile behaviour of in-plane magnetization and longitudinal resistivity versus E -fields shown in Fig. R9b and Fig. R9c [*Phys. Rev. Lett.* **108**, 137203 (2012); *Nat. Nanotechnol.* **14**, 131 (2019)]. The variation of the in-plane magnetization with the E -field guided by the dashed arrows is shown in Fig. R9b, which turns out to be a loop-like (instead of butterflylike) magnetization-electric field response. The in-plane magnetization changes sharply with the polarization switching process in PMN-PT as revealed by

the switching current peak. Figs. R9c and R9d resemble the E -field dependent resistance and gate current I_G in MnPt/PMN-PT heterostructures, and the hysteretic shape is also due to 109° ferroelastic domain switching [Nat. Nanotechnol. **14**, 131 (2019)]. Therefore, 109° ferroelastic switching is important for the nonvolatility in E -field control of in-plane magnetization and longitudinal resistivity, which results in the memory effect in our hybrid FE/FM device.

Fig. R9 Ferroelectric domain switching and looplike magnetization-electric field response for CoFeB/PMN-PT structure, and the electro transport properties of the MnPt film. **a**, Scheme for different distortions caused by $71^\circ/180^\circ$ and 109° polarization switching along $[110]$ and $[-110]$. **b**, E -field tuning of the in-plane magnetization (square) and polarization current (open circle). [Phys. Rev. Lett. **108**, 137203 (2012)] **c**, E -field dependent four-probe resistance of the MnPt film, which exhibits a hysteretic and asymmetric butterfly shape. **d**, E -field dependent gate current of the PMN-PT substrate. The curved arrows and numbers represent the measurement procedure. [Nat. Nanotechnol. **14**, 131 (2019)]

As shown in Fig. R10a, the looplike response of the AHE signal ΔV_{xy} appeared after applying a negative polarized electric-field, indicating that the 109° ferroelastic switching is the dominating factor in our system. And this non-volatile behaviour still remains with sweeping the E -field in positive range (Fig. R10b). The nonvolatility of the AHE signal makes it possible to be an adequate physical reservoir with memory effect. We have added the results of the nonvolatile resistivity change controlled by strain in supplementary information S7.

Fig. R10 Nonvolatility of the AHE signal ΔV_{xy} after applying the negative E -field. **a, E -field between -8.3 to 15 kV/cm. **b**, E -field between 0 to 15 kV/cm after the first cycle shown in **a**.**

Referee's Comment #5:

In Fig. 3b (which seems to be one of the key results), Why the labyrinth domain and the saturated state respond similarly to the input signal? Why the Skyrmion state respond to the input signal more strongly?

Authors' Reply:

We apologize for the unclear description in the manuscript. The Hall resistivity ($\rho_{xy}(E)$) changes with the strain generated by the E -field including two components, one is the perpendicular magnetization change ($M_z(E)$), and the other is the longitudinal resistivity change ($\rho_{xx}(E)$). The empirical formula is written in Eq. (1), $\rho_{xy}(E) = R_0 H_z + R_s 4\pi M_z(E) + a\rho_{xx}(E)$.

In the labyrinth domain and saturated states, the perpendicular magnetizations do not change under the variation of the E -field, where $M_z(E)$ remains zero. So, the Hall resistivity change is only attributed to the longitudinal resistivity change under E -field ($\rho_{xx}(E)$), which is mainly influenced by the ferroelastic strains generated by the E -field [*Nat. Nanotechnol.* **14**, 131 (2019); *J. Appl. Phys.* **127**, 244102 (2020)]. Thus, the longitudinal resistivity change with strain is not related to the magnetization states.

The labyrinth domain and saturated states show almost the same response to the input signal because these 2 magnetization states are less susceptible than the skyrmion state under the applied E -field. We all know that the DMI and anisotropy changed by the E -field could not affect the saturated state. While for the labyrinth domain state, as shown in Fig. R11 [*Nat. Commun.* **11**, 3577 (2020)], the magnetization barely changes with E -fields, either. Therefore, these 2 states respond similarly to the input signal. We deleted the output result of the labyrinth domain in the revised manuscript, so it can help to separate the contribution from $M_z(E)$ and $\rho_{xx}(E)$.

Fig. R11 MFM images under different E -fields for $[\text{Pt}/\text{Co}/\text{Ta}]_{12}$ multilayers without external magnetic field. The scale bar is $1 \mu\text{m}$. [*Nat. Commun.* **11**, 3577 (2020)]

The skyrmion state, which is the nucleation of magnetic domains, is sensitive to the magnetic anisotropy and DMI change induced by strain. Thus, the perpendicular magnetization component varies with the applied E -fields, as illustrated in Fig. 2b and Fig. R12. The Hall resistivity $\rho_{xy}(E)$ is a combination of $M_z(E)$ and $\rho_{xx}(E)$, so the change of magnetic anisotropy and DMI would influence the skyrmion state most among all 3 states. Specifically, the skyrmion deformation plays a vital role, which has been discussed in the revised manuscript *discussion* part. Thus, skyrmion deformation enhances the Hall voltage as output signal, and we named this physical system the skyrmion-enhanced RC.

Fig. R12 MFM images under different E -fields for $[\text{Pt}/\text{Co}/\text{Gd}]_7$ multilayers in the skyrmion state. The MFM image size is $5 \times 5 \mu\text{m}^2$.

Referee's Comment #6:

Third, it seems that the three physical reservoirs the authors mentioned (labyrinth domain, saturated state, and skyrmions) cannot coexist, and that they cannot be switched from one state to another by electric field. Rather, they need to be changed by changing the bias magnetic field. Would that influence the Reservoir computing application?

Authors' Reply:

We would like to thank the referee for the comment. Indeed, the labyrinth domain, saturated state, and skyrmions are three independent reservoirs. We make use of every single state as an independent reservoir and train their output weights one by one. Then we can compare the recognition rate of each reservoir and find the reservoir in skyrmion state showing better performance. So, the three states are separated from each other and it would not influence the RC

application.

Our system is different from the previously reported reservoir in reference [*Sci. Adv.* **8**, eabq5652 (2022)], where different magnetic states are subsections of the reservoir. To avoid misleading the reader, we have revised Fig. 3a.

We have modified the manuscript and have made this part clear “(lines 164-166) *Figure 3a illustrates the schematic of the strain-mediated spintronic RC framework, where the two different magnetization states (skyrmion phase and saturation) serve as different physical reservoirs, presented by MFM images.*”.

Referee’s Comment #7:

Then there are other problems like why a ferrimagnetic multilayer Pt/Co/Gd was used rather than ferromagnetic multilayer. For a serious device design paper that warrants publication in Nature communications, materials design (what materials, how materials parameters affect the device performances) and device design principles should all be clearly articulated.

Authors’ Reply:

We gratefully thank the referee for the instructive comments.

We choose the ferrimagnetic multilayer Pt/Co/Gd due to 2 reasons. First, in magnetic materials, the absolute value of the magnetostriction coefficient of CoGd is relatively large ($\lambda = -1.7 \times 10^{-4}$ for GdCo₂ alloy [*Phys. Stat. Sol. (a)* **34**, 383 (1976)]). It will make the strain-induced anisotropy change greater. Thus, larger amplitude of the output signal can be achieved. Second, the resistivity of Gd is 130 $\mu\Omega\cdot\text{cm}$, which is 1~2 orders of magnitude greater than that of other heavy metals like Ta, W and Ir. Thus, large longitudinal resistance changes under application of *E*-fields will increase the amplitude of the AHE signal ΔV_{xy} , and further improve the signal-to-noise ratio for the output voltages. Therefore, we choose Pt/Co/Gd as magnetic skyrmion layer.

Additionally, there are also some potential advantages. Ferrimagnetic systems exhibit ultrafast magnetization dynamics compared to ferromagnetic system, which can reach sub-THz [*Phys. Rev. B* **100**, 100409(R) (2019)]. Also, the breathing dynamics of skyrmions work at ~GHz, and show great nonlinear responds, which is appealing for ultrafast reservoirs [*Phys. Rev. Appl.* **12**, 024008 (2019); *Neuromorph. Comput. Eng.* **2**, 044011 (2022); Preprint at <https://arxiv.org/abs/2209.06962> (2022)]. Those high frequency properties indicate potential for ultrafast RC application based on ferrimagnetic skyrmions.

We designed the Hall bar device as a prototype, and used a typical electrical measurement method, AHE. The device is designed for all-electrical input and output. For further development, the readout device can be replaced by magnetic tunnel junctions (MTJs) for device miniaturization in future research. MTJs will enhance both magneto- and electro- resistance change [*Nat. Nanotechnol.* **14**, 131 (2019); *Nat. Commun.* **10**, 243 (2019); *Adv. Mater.* **32**, 2002300 (2020); *Sci. Bull.* **67**, 691 (2022); *Adv. Electron. Mater.* **9**, 2200570 (2023)]. Thus, better performance of the skyrmion reservoir than other magnetization states could be expected.

In the revised manuscript, we have modified the description of the device and material design principles “(lines 329-332) *In an effort to realize the skyrmion enhancement, the magnetic tunnel junction (MTJ) can be a replacement of Hall bar device to magnify the output signal induced by*

skyrmion deformation, and make it comparable with the longitudinal resistivity variation with the magnitude of mV [42,49,63].”, and “(lines 351-352) Moreover, ferrimagnetic system exhibits the ultrafast magnetization dynamics compared to ferromagnetic system, which can reach sub-THz [66,70].”.

Referee's General Remarks:

The authors demonstrate reservoir computing (RC) using AHE response to an input voltage applied to a PMN-PT (100) substrate on which Pt/Co/Gd multilayers are deposited. They explore magnetization in the heterostructure film is in saturated, skyrmion and labyrinth states and explain the reason behind the skyrmion state being most effective for RC, which is its being most responsive to the E-field. (This claim for the AHE responsiveness to E-field is corroborated by MFM images under applied E-field in the skyrmion state in the supplement).

They further show that with appropriate training of weights of the output layer of their RC system they can classify waveforms and predict Mackey-Glass time series. Overall, I think this work is an important contribution to the field of physical reservoir computing using spintronics/nanoscale magnetism. However, before it is suitable for publication several important issues need to be addressed.

Authors' Reply:

We gratefully thank the referee for his/her support and positive comment that our work is “an important contribution to the field of physical reservoir computing using spintronics/nanoscale magnetism”. In this work, we demonstrated a full electrical input and output reservoir computing (RC) system with appealing ultra-low-power consumption. The benchmark task of Mackey-Glass time series has been successfully predicted using our RC system, and a NRMSE of 0.2 for a 20-step prediction is achieved. RC is a one kind of recurrent neural network, which is a competent architecture of neuromorphic computing. This work can provide alternatives for energy-efficient skyrmionic neuromorphic computing and offer a guide for high performance strain-mediated spintronic RC systems.

In the revised manuscript, we added the experimental results on the benchmarking task of our reservoir, and also more discussion on its physical origin. With such modifications, we believe that our work is suitable for publication in *Nature Communications*.

Referee's Comment #1:

1. Strain generation by applying an electric field: Figure 1 shows that voltage is applied via a back-gate while a top electrode in the corner of PMN-PT substrate is grounded. Considering PMN-PT is an insulator how can there be field lines perpendicular to the PMN-PT substrate unless the whole top surface is grounded. (If a complete top electrode is applied and grounded it would not allow the transport/AHE measurements).

Authors' Reply:

We would like to thank the referee for this valuable comment. The back-gate is a full-film of Cr(20 nm)/Au(100 nm) deposited on the bottom of the PMN-PT substrate, while the top electrode is 1 mm×1 mm conductive silver paint in the corner of the sample. Besides, the nanovoltmeter and

current source meter (connect to the Hall bar device) are grounded together with the high voltage source which provides the input signal across the PMN-PT substrate, as shown in Fig. 1. The voltage applied along the Hall bar for the output signal is $\Delta V_{xx} = \Delta R_{xx} \times I = 33.4 \Omega \times 500 \mu\text{A} = 16.7 \text{ mV}$, which is 4 orders of magnitude lower than the applied voltage across to the PMN-PT substrate (0 to 450 V for E -field 0 to 15 kV/cm). Therefore, the whole top surface of PMN-PT can be considered as being on the same ground potential, where the electric field lines are perpendicular to the top surface.

Referee's Comment #2:

2. One possibility is that the magnetic state is also sensitive to small magnetic fields (in addition to strain). Did the authors ensure that is the process of applying a time varying voltage to the PMN-PT substrate, they did not also produce a small magnetic field and the RC was a result of this and not the E -field.

Authors' Reply:

We sincerely thank the referee for this valuable comment. In order to verify the Hall signal change in our work is due to the strain, rather than small magnetic fields produced by the time varying voltage, we performed the same experimental procedure using Pt/Co/Gd multilayers deposited on a Si substrate for comparison.

First, the same Hall bar device was fabricated on the Si substrate sample, whose multilayers were prepared at the same time together with the sample on the PMN-PT substrate. Second, the E -field was applied across the sample, where a 0.15 mm insulating glass substrate is glued by silver paint at the bottom of the Si substrate. The thin glass substrate is used to endure the comparable E -field applied on PMN-PT substrate (0 to 450 V for E -field 0 to 30 kV/cm), since the resistance of this Si substrate is around $\sim\text{k}\Omega$. We then measured the AHE loop of the device, as shown in Fig. R13a. The same behaviors are observed, indicating the magnetization does not change with applied E -field, because the Si substrate does not have a piezoelectric effect.

After that, we performed the same waveform recognition task and the result is shown in Fig. R13b. All 3 magnetization states show random noise-like output results with a smaller amplitude (0.5 μV) compared with the result in the article (3 μV for skyrmion state and 1.5 μV for the saturation state). These results prove that the small magnetic field produced by the time varying voltage did not show influence on the magnetic states, i.e., the output AHE signal V_{xy} . Therefore, the recognition result shown in Pt/Co/Gd multilayers on PMN-PT is coming from the strain instead of the small magnetic field produced by the time varying voltage.

Fig. R13 Same experimental procedure performed using the Hall bar device on Si substrate. **a**, AHE loops under different E -fields. The 3 arrows denote the 3 magnetization states in **b**. **b**, Input waveform sequences and the corresponding output signals of the device on Si substrate. The blue, red, and grey output signals correspond to the labyrinth domain, skyrmion, and saturation states, respectively.

Referee's Comment #3:

1. I do not see how the Neural ODE adds any value to the key message of the paper, which is demonstrating that the AHE output due to the voltage induced strain can implement RC effectively in the skyrmion state. Whether the AHE (magnetization) response can be predicted/simulated by Neural ODE/micromagnetic does not prove/disprove the efficacy of this system for RC.

Authors' Reply:

We thank the referee for pointing out this question. In the original manuscript, the waveform classification task has already proved the enhancement of skyrmion state in our strain-mediated reservoir (in terms of the recognition rate). When studying a more complex task like the MG time series prediction task, the noise of the measurement system is on a sub μV level, which submerges the small signal information since the peak-to-peak signal is a few μV . In this case, we took advantage of the Neural ODE to build an ideal (noiseless) model for our nonlinear system to perform the complex MG task, in order to give an instruction for the experiments. By now we have increased the output signal with a magnitude of mV through controlling the direction, duration time, and amplitude of the E -field applying to the PMN-PT substrate for initial polarization (which can increase the proportion of 109° ferroelastic domain switching [*Adv. Mater.* **33**, 2103013 (2021)]) and it is large enough to perform the MG prediction task against the system noise, and the result is displayed in Fig. R14 (the same figure in the revised manuscript as Fig. 4). Additionally, the prediction performance of the experiment behaves similar with the constructed model by Neural ODE, as compared in Fig. R15.

We perform the MG task experimentally and added this to the revised manuscript. The pre-processed input signal and the output V_{xy} are shown in Fig. R14a (the grey and the red data sets, respectively). Utilizing the nonlinearity and short-term memory of the strain-mediated system, the predicting results in terms of the prediction error is evaluated by a normalized root mean square error (NRMSE), which can reach 0.2 for a 20-step prediction, as shown in Fig. R14b.

Figures R14c to R14f show the predicting results of the MG time series for the next value ($i = 1$), and the value happening 20 steps later ($i = 20$), using the basic method and the optimized algorithm (extending the reservoir size by using previous steps' output states) for output reconstruction. Each horizontal prediction step i corresponds to a different prediction task. In the figures, the blue curve is the ground truth, the black and the red curves are prediction results from basic and optimized methods, respectively. The prediction error apparently increases in Fig. R14d compared to R14c, and also the same trend shown in Fig. R14f compared to R14e. Additionally, the predicted results using the optimized algorithm shown in Figs. R14e and R14f demonstrate better accuracy than those using the basic method shown in Figs. R14c and R14d, respectively. To be specific, the NRMSE values are 0.16 and 0.2 for predictions of $i = 1$ and $i = 20$ using the optimized method, correspondingly the values are 0.27 and 0.34 using the basic method for output reconstruction. The results indicate the great potential for our physical reservoir with extended reservoir size.

Fig. R14 Mackey-Glass time series prediction task demonstrated by the strain-mediated spintronic RC system. **a**, Input pre-processed signal (grey data set) and the corresponding output signal (red data set) of the strain-mediated reservoir. **b**, NRMSE as a function of horizontal prediction step i , shown in log scale, for the testing set by using the basic method (black curve) and the optimized algorithm (red curve) for output reconstruction. **c** and **d** are the selected predicting results (black) for horizontal step $i = 1$ (for short-term prediction) and 20 (for long-term prediction), respectively, predicted by the strain-mediated RC system using basic method for output reconstruction, in comparison with the ground truth (blue) of MG time series. **e** and **f** are the selected predicting (red) results for horizontal step $i = 1$ and 20, respectively, predicted by the strain-mediated RC system using optimized algorithm for output reconstruction, in comparison with the ground truth (blue) of MG time series.

Fig. R15 MG time series prediction task performance of the strain-mediated spintronic RC system and the constructed model. The black and red curves are calculated from experimental data (the same with Fig. R14b). And the blue curve and the green dashed curve are the result of constructed models by Neural ODE for skyrmion and saturated states, respectively.

We have revised the manuscript “**Strain-mediated RC System for Mackey-Glass time series prediction task**” part (lines 213-275), and the supplementary information S6.

Referee’s Comment #4:

2. With E field perpendicular to the substrate and along the poling direction, the PMN-PT (100) substrate would be isotropic in-plane. Hence, the effect of strain is modulating the PMA. Could the authors clarify this?

Authors’ Reply:

We gratefully thank the referee for the constructive suggestion. We evaluate the magnetic anisotropy changes under E-fields, as shown in Fig. R16.

MOKE with rotating magnetic fields (Rot-MOKE) was conducted to evaluate the effective magnetic anisotropy fields quantitatively with the applied E-fields. For Rot-MOKE experiments, we started rotating from the polar-MOKE configuration as illustrated in Fig. R16a. Figure R16b shows the experimental (data points) and fitting (curves) results of the torque $l(\theta)$ as a function of the magnetization equilibrium angle θ under different applied E-fields. The effective magnetic anisotropy fields ($H_{k,eff}$) are -267 , -504 and -731 Oe for 0 , -10 and -20 kV/cm, respectively. A negative $H_{k,eff}$ is obtained, which means that there is an in-plane magnetic anisotropy in Pt/Co/Gd multilayers. The in-plane magnetic anisotropy presumably originates from the strong dipolar interaction due to the multi-repetitions, since the trilayer of Pt/Co/Gd shows clear out-of-plane easy axis loop in Fig. R16c. The change of magnetic anisotropy fields $\Delta H_k(E) = H_{k,eff}(E) - H_{k,eff}(E_0)$ is plotted as a function of the sweeping E-fields, shown in Fig. R16d. A butterfly shape (typical strain-E relationship of PMN-PT substrates) with the same switching fields (± 1.67 kV/cm) is shown

in Fig. 2b, meaning that the anisotropy change originates from the strain generated by E -fields.

Here, the effective magnetic anisotropy is $K_{\text{eff}} = K_u - \frac{\mu_0 M_s^2}{2} = \frac{\mu_0 M_s H_{k,\text{eff}}}{2}$, where K_u is the uniaxial magnetic anisotropy, μ_0 is vacuum permeability and M_s is the saturation magnetization. Utilizing the strain-mediated magnetoelectric coupling in the piezoelectric/ferrimagnetic heterostructure, the simplified strain-induced anisotropy is written as $K_e = -\frac{3}{2}\lambda\sigma$, $\sigma = \varepsilon E_f / (1 - \nu^2)$, where λ is the magnetostriction coefficient, ε is the strain, E_f is the Young's modulus, and ν is the Poisson ratio [*Phys. Rev. B* **84**, 012404 (2011)]. For Pt/Co/Gd multilayers, we took $E_f = 168, 209, \text{ and } 55$ GPa for Pt, Co and Gd to get the effective Young's modulus, $\nu = 0.3$ for metals and $\lambda = -1.7 \times 10^{-4}$ for GdCo₂ alloy [*Phys. Stat. Sol. (a)* **34**, 383 (1976); <https://periodictable.com/Properties/A/YoungModulus.html>]. For PMN-PT (100) single crystal, $\varepsilon = 0.2\%$ at ± 20 kV/cm was taken [*Appl. Phys. Lett.* **78**, 2551 (2001)]. The strain-induced anisotropy change in Pt/Co/Gd multilayer can be estimated: $K_e = 8.5 \times 10^4$ J/m³. The equivalent anisotropy field change is $\Delta H_e = \frac{2K_e}{\mu_0 M_s} = 1416.7$ Oe, where the measured $M_s = 1.2 \times 10^6$ A/m. The measured $\Delta H_{k,\text{eff}} = 464$ Oe (between 0 and -20 kV/cm) is lower than the estimated value, which is presumably due to the transferred strain loss in the multilayers compared to the PMN-PT substrate. Additionally, for the positive piezoelectric coefficients d_{33} in PMN-PT, a tensile strain in out-of-plane direction is generated when the E -field is higher than the ferroelectric switching field, resulting in an in-plane compressive strain. Considering the strain transferred from PMN-PT to the multilayers on top, together with the negative λ in CoGd, a decrease of the out-of-plane magnetic anisotropy can be concluded. Namely, the in-plane magnetic anisotropy increases with E -fields, which is consistent with the measured results, as the negative ΔH_k shown in Fig. R16d.

Fig. R16 Evaluation of magnetic anisotropy changes with E -fields. **a**, Sketch of Rot-MOKE measurements. **b**, The torque $l(\theta)$ versus θ with different E -fields, the solid curves are fitting results. **c**, Out-of-plane hysteresis loop for the Pt/Co/Gd trilayer sample, where the thickness of the Co layer is 1.95 nm. **d**, Fitting results of the magnetic anisotropy field change ΔH_k as a function of the E -fields. The arrows indicate the E -field sweeping direction.

We have revised the manuscript with an additional *discussion* part including the analysis of the strain-induced PMA change (lines 307-310) and added the results of anisotropy change versus strain in supplementary information S4.

Referee's General Remarks:

The primary contributions of this manuscript are:

(1) the first experimental demonstration of a reservoir computer based on nanomagnetic textures with electrical inputs; and

(2) the first simulation of a skyrmion reservoir computer performing a time-series forecasting task.

This manuscript also includes:

(3) analysis of the experimental and simulation results, including an estimate of energy dissipation.

Authors' Reply:

We gratefully thank the referee for positive comments on the contributions of our work. In this work, we have demonstrated a full electrical input and output reservoir computing (RC) system with appealing ultra-low-power and multi-parameter fusion. Following the suggestions of the referees, in the revised manuscript, we have added experimental demonstrations of RC benchmarking task, and also, we have added some discussion on the physical mechanism of our reservoir to analyze the experimental and simulation results as well as the estimation of energy dissipation.

Therefore, we have pushed the contributions the referee listed a step further, which are:

- (1) For the experimental demonstration of the waveform recognition task, we further calculated the parity check capacity and short-term memory content to evaluate the performance of our reservoir quantitatively.
- (2) Besides the simulation of skyrmion-based reservoir to perform the time series forecasting task, experimental demonstration of benchmarking Mackey-Glass time series prediction task has been realized and a NRMSE of 0.2 for a 20-step prediction is achieved.
- (3) The physical insights of the reservoir have also been discussed, including the role of magnetic textures, the functionality of ferroelectric materials and the strain-induced magnetic anisotropy changes. We also have discussed the energy dissipation of the strain-mediated spintronic RC system in detail.

Referee's Comment #1:

(1) The experiment convincingly demonstrates that the skyrmion reservoir computer indeed functions as a reservoir. However, this reservoir seems to function considerably slower and provide reduced computational capabilities relative to alternative reservoirs. To enable evaluation of this reservoir's computational capabilities, the authors should report its parity check capacity and short-term memory content; this calculation may be performed through further analysis of the experimental data that has already been presented.

Authors' Reply:

We sincerely thank the referee for the valuable suggestions. We evaluate the strain-mediated spintronic reservoir's computational capabilities through the parity check (PC) capacity and short-

term memory (STM) content [*Phys. Rev. Appl.* **10**, 034063 (2018)] by further analyzing the experimental data of the waveform classification task. The result is shown in Fig. R17. $T_{\text{delay, max}} = 2$ is used in this calculation, due to the experimental results of the waveform classification only comprise the current, the last, and the second-to-last waveform. We define C_{STM} as the capacity for short-term memory and C_{PC} as the capacity for PC. The capacity at skyrmion state is the largest among three magnetization states, which is in accordance with the recognition rate results shown in the manuscript.

As the referee pointed out, our reservoir functions slower and holding smaller memory capacity than other spintronic reservoirs, according to the waveform recognition task results. Limited to the data acquisition setup, the period of the waveform is in the time scale of seconds. The short-term memory of magnetic dynamics, such as magnetic after-effects (normally in ms scale), cannot be reflected in our reservoir system.

However, we make a prospect on skyrmion RC potentially works at GHz based on these reasons: First, the piezoelectric switching and ferroelectric polarization switching time can be extremely fast and less than 1 ns [*Appl. Phys. Lett.* **89**, 021109 (2006); *IEEE Int. Electron Devices Meeting (IEDM)* 15.2.1 (2019)]. Second, ferrimagnetic systems exhibit ultrafast magnetization dynamics compared to ferromagnetic system, which can reach sub-THz [*Phys. Rev. B* **100**, 100409(R) (2019)]. Third, our reservoir operates with magnetic skyrmions, where the breathing mode frequency of skyrmion is in the order of GHz. For example, Chen, X., et al. studied the characteristic frequency of the skyrmion breathing can reach above 7 GHz [*Phys. Rev. Appl.* **12**, 024008 (2019)]. With the adaption of the breathing of skyrmions, the frequency of this skyrmion based RC can be increased, and the recent work by Rajib, M.M., et al. is a good example [*Neuromorph. Comput. Eng.* **2**, 044011 (2022)]. Recently, a task-adaptive approach has been researched in a chiral magnet Cu_2OSeO_3 that host different magnetic phases including skyrmion, helical and conical, which utilizes the GHz spin dynamics of the chiral magnet [Preprint at <https://arxiv.org/abs/2209.06962> (2022)]. Those high frequency properties indicate the potential for ultrafast RC application based on ferrimagnetic skyrmions.

We have revised the manuscript with an additional *discussion* part including the potential of the strain-mediated spintronic reservoir to work at gigahertz (lines 346-352).

Fig. R17 Calculated correlation and capacity results of STM and PC. **a** and **b**, The correlation using the method in [*Phys. Rev. Appl.* **10**, 034063 (2018)]. The integrated values of **a** are defined as the short-term memory capacity (C_{STM}), and the integrated values of **b** are defined as the parity check capacity (C_{PC}). **c**, C_{STM} and C_{PC} for labyrinth domain, skyrmion (both the experiment and the Neural ODE model), and saturation states.

Referee's Comment #2:

(2) The authors have used the neural ODE approach to demonstrate that their system can implement the Mackey-Glass time series prediction task; however, while this neural ODE approach can give design guidance and provide a rough estimate of how it will work IF it works, it cannot be considered a reliable and authoritative prediction of system functionality in a manner similar to micromagnetic simulations. Therefore, the authors cannot rely solely on this neural ODE simulation in order to prove that this reservoir computer can perform a time-series forecasting task. As their experimental setup appears suitable for demonstration of the Mackey-Glass task, why didn't the authors perform this task experimentally?

Authors' Reply:

Thank you very much for pointing out this question. In the original manuscript, the waveform classification task has already proven the enhancement of the skyrmion state in our strain-mediated reservoir (in terms of the recognition rate). When studying a more complex task like the MG time series prediction task, the noise of the measurement system is on a sub μV level, which submerges the small signal information since the peak-to-peak signal is a few μV . In this case, we took advantage of the Neural ODE [*Nat. Commun.* **13**, 1016 (2022)] to build an ideal (noiseless) model for our nonlinear system to perform the complex MG task, in order to give an instruction for the experiments.

By now we have increased the output signal to a magnitude of mV through controlling the direction, duration time, and amplitude of the E -field applying to the PMN-PT substrate for initial polarization (which can increase the proportion of 109° ferroelastic domain switching [*Adv. Mater.* **33**, 2103013 (2021)]) and it is large enough to perform the MG prediction task against the system noise, and the result is displayed in Fig. R18 (the same figure in the revised manuscript as Fig. 4). Additionally, the prediction performance of the experiment behaves similarly with the constructed model by Neural ODE, as compared in Fig. R19.

We perform the MG task experimentally and added this information to the revised manuscript. The pre-processed input signal and the output V_{xy} is shown in Fig. R18a (the grey and the red data sets, respectively). Utilizing the nonlinearity and short-term memory of the strain-mediated system, the predicting results in terms of the prediction error is evaluated by a normalized root mean square error (NRMSE), which can reach 0.2 for a 20-step prediction, as shown in Fig. R18b.

Figures R18c to R18f show the predicting results of MG time series for the next value ($i = 1$), and the value happening 20 steps later ($i = 20$), using the basic method and the optimized algorithm (extending the reservoir size by using previous steps' output states) for output reconstruction. Each horizontal prediction step i corresponds to a different prediction task. In the figures, the blue curve is the ground truth, the black and the red curves are prediction results from basic and optimized methods, respectively. The prediction error apparently increases in Fig. R18d compared to R18c, and also the same trend shown in Fig. R18f compared to R18e. Additionally, the predicted results using the optimized algorithm shown in Figs. R18e and R18f demonstrate better accuracy than those using the basic method shown in Figs. R18c and R18d, respectively. To be specific, the NRMSE values are 0.16 and 0.2 for predictions of $i = 1$ and $i = 20$ using the optimized method, correspondingly the values are 0.27 and 0.34 using the basic method for output reconstruction. The results indicate the great potential for our physical reservoir with extended reservoir size.

Fig. R18 Mackey-Glass time series prediction task demonstrated by the strain-mediated spintronic RC system. **a**, Input pre-processed signal (grey data set) and the corresponding output signal (red data set) of the strain-mediated reservoir. **b**, NRMSE as a function of horizontal prediction step i , shown in log scale, for the testing set by using the basic method (black curve) and the optimized algorithm (red curve) for output reconstruction. **c** and **d** are the selected predicting results (black) for horizontal step $i = 1$ (for short-term prediction) and 20 (for long-term prediction), respectively, predicted by the strain-mediated RC system using basic method for output reconstruction, in comparison with the ground truth (blue) of MG time series. **e** and **f** are the selected predicting (red) results for horizontal step $i = 1$ and 20, respectively, predicted by the strain-mediated RC system using optimized algorithm for output reconstruction, in comparison with the ground truth (blue) of MG time series.

Fig. R19 MG time series prediction task performance of the strain-mediated spintronic RC system and the constructed model. The black and red curves are

calculated from experimental data (the same with Fig. R18b). And the blue curve and the green dashed curve are the result of constructed models by Neural ODE for skyrmion and saturated states, respectively.

We have revised the manuscript “**Strain-mediated RC System for Mackey-Glass time series prediction task**” part (lines 213-275), and the supplementary information S6.

Referee’s Comment #3:

(3) The authors estimate that the energy dissipation for the waveform classification task is less than 1 fJ per waveform. However, this estimate does not include the energy required to read the output. In many physical reservoir proposals, the output read energy is far larger than the input energy; I expect this to be the case for the authors’ proposed system as well. It is therefore critical that the authors consider the output read energy, which will likely dominate the total energy dissipation estimate.

Authors’ Reply:

We would like to thank the referee for this valuable comment. We now add the readout energy into the total energy dissipation for our RC system in the revised manuscript *discussion* part (lines 341-345), and the supplementary information S8.

In the AHE measurements, a DC current of $I_{dc} = 0.5$ mA, and the output V_{xy} is less than 50 mV. The Joule heating energy on the Hall bar device is 25 μ W. When decreasing the device size into a 100 nm diameter nanomagnet, the cross-area of the Hall bar will be $(100 \text{ nm})^2/10 \mu\text{m} \times 150 \mu\text{m} \approx 1/150000$ of the original size. The energy dissipation will reduce to 1/150000 of 25 μ W, i.e., 166.7 pW. Considering the 1 GHz waveform signal as discussed in the supplementary information S8, the power dissipation is much less than the estimated 850 nW. Therefore, the energy dissipation for the waveform classification task is still less than 1 fJ per waveform.

Referee's General Remarks:

This paper reports the strain enhancement due to the formation of magnetic skyrmions in PMN-PT. They performed benchmarking using Mackey-Glass series. This can be useful for reservoir computation with low-power and high-speed operation as the authors pointed out. This paper can be published in Nat. Commun. after considering the following issues.

Authors' Reply:

We greatly appreciate the referee's positive comment on our paper as "*This can be useful for reservoir computation with low-power and high-speed operation as the authors pointed out*". And we have improved our manuscript according to the referee's comments and suggestions. In this work, we demonstrated the full electrical input and output reservoir computing (RC) mediated by strain, with appealing ultra-low-power and multi-parameter fusion. In the revised manuscript, the benchmarking Mackey-Glass time series prediction task is experimentally performed and a NRMSE of 0.2 for a 20-step prediction is achieved. In the view of energy dissipation, our physical reservoir shows a 2 orders of magnitude reduction of energy dissipation than other spintronic reservoirs. Additionally, the role of spin textures such as magnetic skyrmions in our device is discussed in detail to give the physical insights on the skyrmion enhancement of the performance, which is shown in the recognition rate of the waveform recognition task. This work is significant for energy-efficient strain-mediated neuromorphic computing and we believe that the revised manuscript is suitable for publication in *Nature Communications*.

Referee's Comment #1:

(1) The authors used the term of "straintronics" throughout their manuscript. Unfortunately this term is not commonly used in the community and is needed some clear definition in the introduction.

Authors' Reply:

We would like to thank the referee for this instructive comment. In order to make it clear, we have revised "straintronic" to "strain-mediated spintronic", because the strain-induced physical effects are mainly studied in our physical reservoir devices.

Referee's Comment #2:

(2) In Fig. 1, the meaning of the blue "Straintronic reservoir" diagram is not clear. The authors should also include the dimensions of their device in Fig. 1. They may wish to include an optical micrograph or scanning electron micrograph of their device in their supplemental information. They should also explain the width and amplitude of the input signals in the caption. They should provide a clear MFM image with a scale bar showing the magnetic force in their supplement.

Authors' Reply:

We sincerely thank the referee for these valuable comments. We have revised Fig. 1 as the referee suggested, as shown in Fig. R20a. The dimension of our Hall bar is $150 \times 900 \mu\text{m}^2$, which has been added to the caption of Fig. 1. The original optical micrograph of the device is shown in Fig. R20b. The MFM image with a color bar showing the magnetic force is presented in Fig. R20c (the same figure as Fig. 5a in the revised manuscript). The detailed information about the input signal (*the width and amplitude*) can be found in Fig. 3b and Fig. 4a.

Fig. R20 a, Revised Fig. 1. **b**, Original optical micrograph of the device shown in Fig. 1, the scale bar is $200 \mu\text{m}$. **c**, MFM image with the color bar showing the magnetic force (phase shift), the scale bar is $1 \mu\text{m}$.

Referee's Comment #3:

(3) The authors should add the details of the quality of their devices. Especially they should add a cross-sectional transmission electron micrograph to confirm the quality of the interfaces, which are crucial to determine the skyrmion formation.

Authors' Reply:

We gratefully thank the referee for this constructive suggestion. We add a cross-sectional transmission electron micrograph (TEM) to confirm the quality of the interfaces as shown in Fig. R21a, with the sample structure of Si/Ta (2 nm)/Pt (3 nm)/[Co (1.95 nm)/Gd (1.2 nm)/Pt (3 nm)]₇. The multilayers are polycrystalline with sharp interfaces can be seen in the high angle annular dark-field scanning transmission electron microscopy (HAADF-STEM) image together with the energy dispersive X-ray (EDX) mapping shown in Figs. R21b and R21c. The periodic Pt/Co/Gd multilayer structure further is also confirmed in Fig. R21c. The Pt/Co interface is sharper than Gd/Pt, which is mainly because the Gd layer is very thin (1.2 nm) and also Gd atoms can easily diffuse. These distinct interfaces ensure the formation of the skyrmions.

Fig. R21 Characterization of the sample's structure. **a**, Cross-sectional TEM image of the multilayer. **b**, Cross-sectional HAADF-STEM image of the multilayer. **c**, Corresponding EDX mapping for element distributions of Pt (green), Co (red), Gd (blue) and Ta (magenta). The scale bar is 5 nm for **a**, **b** and **c**.

We have added these results in supplementary information S1.

Referee's Comment #4:

(4) In Fig. 2a, the authors showed two MFM images as insets. Were they taken at the same position? It would be better to extract the density of the skyrmions from these images to support their discussion as well as the size distributions of the skyrmions under different magnetic fields.

Authors' Reply:

We gratefully thank the referee for this valuable comment.

The MFM images shown in Fig. 2a are taken at the same position. MFM measurements are performed in the tapping/lift mode, thus the sample morphology and magnetic structures are obtained at the same time. Two MFM images in Fig. 2a and their corresponding topography images are shown in Fig. R22. Figures R22a and R22b are taken at $\mu_0 H_a = 90$ mT (the skyrmion state) under the E -field of 0 and 10 kV/cm, respectively. The topography shown in Fig. R22d is same as Fig. R22c with left and downward drift, which can be determined from the defect at the up-right corner (marked by a red circle). Thus, the MFM images in Fig. 2a were taken at the same location on the sample. Additionally, since the same MFM probe was used, and with the same lifted height in these measurements, the stray fields from the MFM probe should be the same.

Fig. R22 MFM images and their corresponding topography images. **a** and **b** are the inset MFM images in Fig. 2a. **c** and **d** are the corresponding topography images. Red circles mark the same defect point of the sample. The image size is $5 \times 5 \mu\text{m}^2$.

According to the density of skyrmions in these images. We binarized the MFM images to count the number of skyrmions as shown in Fig. R23. There are ~ 7 and ~ 115 skyrmions shown in Figs. R23a and R23b, and the corresponding $\Delta M_{\text{skyrmion}}/M_s$ are 1.3% and 13% summarized from the black area, respectively. The $\Delta M(E)$ for E -field between 0 and 10 kV/cm is 11.7%. This result is in a good agreement with the results shown in Fig. 2b (M/M_s extracted from the hysteresis loops measured by MOKE in Fig. 2a), which is $\frac{92\% - 66\%}{2} = 13\%$. The increase of skyrmion density indicates the E -field induced skyrmion generation. We have added these results in supplementary information S3.

Fig. R23 Binarized results of Figs. R22a and R22b. The image size is $5 \times 5 \mu\text{m}^2$.

The size distribution of skyrmions under different magnetic fields is studied by Lorentz transmission electron microscopy (L-TEM). The same sample deposited on a 50-nm-thick Si_3N_4 membrane was used for L-TEM measurements, with a tilting angle of 10° , as shown in Fig. R24. With increasing the magnetic field, the labyrinth domains shrink to magnetic skyrmions, and finally to a fully saturated single-domain state. Figures R24a and R24b show the L-TEM images (the skyrmion state) taken at $\mu_0 H_a = 200$ and 240 mT, respectively. We measured the skyrmion size from 203 mT up to 263 mT and the result is shown in Fig. R24c. The error bar is given by the uncertainty

in distinguish the edges of skyrmions. Below 203 mT and above 263 mT, there are no or just very few skyrmions visible. The skyrmion size distribution is constant within the observed range, showing no trend with the increasing magnetic fields. Note that the magnetic field is different from other measurements maybe due to different substrate (silicon nitride membrane).

Fig. R24 Skyrmion size distributions extracted from L-TEM images. **a** and **b** are the L-TEM images acquired on increasing the magnetic field, $\mu_0 H_a = 200$ and 240 mT, respectively. The image size of **a** and **b** is $5 \times 5 \mu\text{m}^2$. **c**, Skyrmion size distribution statistic results.

Referee's Comment #5:

(5) The authors should comment on the origin of the minor hysteretic behaviour, i.e., two peaks near the zero electric field under the magnetic field application of 90 mT, observed in Fig. 2b.

Authors' Reply:

We thank the referee for this valuable comment. Two peaks near zero electric-field are related to the ferroelectric switching fields of PMN-PT substrate, which is consistent with the variation of the piezoelectric strain versus the applied E -field with ferroelectric switching fields of ± 1.67 kV/cm [*Syst. Struct.* **17**, 15 (2006); *Phys. Rev. B* **75**, 054408 (2007)]. Both the remnant magnetization shown in Fig. 2b and the magnetic anisotropy measured by Rot-MOKE show the typical butterfly shape with the sweeping E -field, indicating that the strain-induced magnetic anisotropy change in this system is due to the magnetoelastic effect. As shown in Fig. R25, the current-voltage (I - V) characteristics of the PMN-PT substrate show 2 peaks at the same E -fields as the peaks in Fig. 2b, which has been discussed in the revised supplementary information S4.

Fig. R25 The I - V curve and the enlarged Fig. 2b.

Referee's Comment #6:

(6) It would be helpful for the readers to include a schematic diagram to show the lattice stretch with % estimated under representative electric fields. Please also discuss the reproducibility and any fatigue to be introduced after a certain operation cycle.

Authors' Reply:

We gratefully thank the referee for the instructive comments. In the revised manuscript, we have added a schematic graph in Fig. 1 bottom panel (also as shown in Fig. R20a). The dashed square and the scaled-down image inside illustrate the in-plane compressive strain when applying the E -field larger than the ferroelectric switching field. Based on our magnetic anisotropy results, the anisotropy field change is $\Delta H_{k, \text{eff}} = 464$ Oe (between 0 and -20 kV/cm). Taking the measured $M_s = 1.2 \times 10^6$ A/m, we can get the effective magnetic anisotropy change: $\Delta K_{\text{eff}} = 2.78 \times 10^4$ J/m³, which is induced by strain. The simplified strain-induced anisotropy is written as $K_e = -\frac{3}{2}\lambda\sigma$, $\sigma = \varepsilon E_f / (1 - \nu^2)$, where λ is the magnetostriction coefficient, ε is the strain, E_f is the Young's modulus, and ν is the Poisson ratio [*Phys. Rev. B* **84**, 012404 (2011)]. For Pt/Co/Gd multilayers, we took $E_f = 168, 209, \text{ and } 55$ GPa for Pt, Co and Gd to get the effective Young's modulus, $\nu = 0.3$ for metals and $\lambda = -1.7 \times 10^{-4}$ for GdCo₂ alloy [<https://periodictable.com/Properties/A/YoungModulus.html>; *Phys. Stat. Sol. (a)* **34**, 383 (1976)]. We estimate the vertical strain generated in our system $\varepsilon_{\text{out-of-plane}}$ is about 0.07%. Assuming approximate volume conservation in the multilayers, we can get the corresponding in-plane compressive strain is $\varepsilon_{\text{in-plane}} = -0.035\%$. The estimated $\varepsilon_{\text{in-plane}}$ is in a reasonable range compared with other works [*Appl. Phys. Lett.* **87**, 262502 (2005); *Phys. Rev. B* **75**, 054408 (2007)].

During the Mackey-Glass time series prediction task, a large amount of data set (grey data set) was sent into the system (around 2500 points \times 50 neurons \times 2 repetitions = 2.5×10^5 points per single

test). As shown in Fig. R26a, the output V_{xy} (red data set) nearly unchanged from beginning to the end in a single test, guided by a pink line. Figure R26b shows the data details of blue rectangle parts in Fig. R26a, and we input 10 data points per second. We tested for more than 100 times in 2 months. The total input (read/write cycle) amount is 2.5×10^7 for unipolar E-field from 0 to 5 kV/cm, which is comparable to the fatigue test studied by Luo, W.G., et al. [*Proc. IEEE Ultrason. Symp.* **2**, 1009 (1999)]. As shown in Figs. R26c and R26d, a loss of 12% in the polarization was observed in the switching cycles up to 2.5×10^8 reversals under bipolar pulse of 12 kV/cm.

Fig. R26 Fatigue test. **a**, Input (grey) and output (red) data sets acquired from one single test of MG prediction task in our experiment. The data of blue rectangles are expanded shown in **b**. **c**, Fatigue behavior of the PMN-PT crystal at bipolar pulses field of 12 kV/cm. **d**, Corresponding hysteresis loops before and after cycling to 2.5×10^8 reversals. [*Proc. IEEE Ultrason. Symp.* **2**, 1009 (1999)]

Referee's Comment #7:

(7) The authors should quantify the statement on line 151, "... roughly ..."

Authors' Reply:

We appreciate the referee's comment. To make it clear, we add some description of the data behavior in the revised manuscript "(lines 173-175) *The behaviour of the output signal coincided with the nonlinearity shown in Fig. 2c, and obvious differences can be seen around 0 kV/cm with*

single and double peaks for square and sine waveforms, respectively.”.

Referee’s Comment #8:

(8) Would the authors be able to change the numbers of skyrmions to be generated monotonically?

Authors’ Reply:

We would like to thank the referee for this comment. In this work, the skyrmions can be changed by both magnetic fields and electric fields, who can change the numbers of skyrmions monotonically.

For the magnetic fields, when decreasing the magnetic field from the saturation state, skyrmions are generated monotonically as shown in Figs. R27a and R27b. Additionally, when increasing the magnetic field from 0 (labyrinth domain), skyrmions can be generated from stripe domains as shown in Figs. R24a and R24b in the response to “Referee’s Comment #4”.

For the electric fields, as can be seen in Fig. 2a state I (Fig. R27c) (low-density skyrmion phase), when increasing the E -field, skyrmions are generated to state II (Fig. R27d) (high-density skyrmion phase). As discussed in the response to “Referee’s Comment #4”, the skyrmions density increase gives rise to the magnetization decrease in the MOKE measurement. The $\Delta M(E)$ (extracted from the MOKE results in Fig. R27e) shown in Fig. R27f represents the skyrmion generation with increasing the E -field monotonically.

Fig. R27 MFM images at different magnetic fields and E -fields. **a** and **b** are the MFM images when decreasing the magnetic field from saturated magnetization. **c** and **d** are the inset MFM images in Fig. 2a, they show increased skyrmion density when increase the E -field. The MFM image size is $5 \times 5 \mu\text{m}^2$. **e**, Out-of-plane magnetic hysteresis loops under different E -fields. The bottom inset shows the enlarged loops around $\mu_0 H_a$, where the skyrmions generated monotonically. **f**, M/M_s extracted from **e** versus the E -field. The arrows indicate the E -field scanning directions.

Referee’s Comment #9:

(9) On line 178, the authors should clarify the meaning of "the short-term memory". What is the time scale they envision?

Authors' Reply:

We thank the referee for this constructive suggestion. The “short-term memory” in the waveform recognition task, means recognition ability for the last and second-to last waveforms, not the absolute time. From the result shown in Fig. 3d, we can draw the conclusion that, the physical reservoir could remember the last waveform precisely (99.3% recognition rate in skyrmion state) and barely remember the second-to last waveform (recognition rate around 50%). The time scale we refer to “the short-term memory” is 2-waveform duration time. Specifically, to our experiment, the time scale is about $9\text{ s} \times 2 = 18\text{ s}$. In the revised manuscript, we have clarified the time scale as “(lines 198-200) *This result demonstrates the short-term memory effect of the strain-mediated reservoir, which is 2-waveform duration time (specifically to our experiment, the time scale is about $9\text{ s} \times 2 = 18\text{ s}$).*”.

Referee's Comment #10:

(10) It might not be so clear for the readers how the authors would use the skyrmions for reservoir computation as schematically shown in Fig. 3a. How advantageous would their device be as compared with the other concepts for the reservoir computation, such as in terms of their operation range?

Authors' Reply:

We apologize for the misleading in Fig. 3a. Actually, in this work, each of the magnetic states (magnetic skyrmions and saturation state) can solely operate as a physical reservoir. And based on this fact, we have compared the performance of these two independent reservoirs, where skyrmion exhibits better performance. We have modified Fig. 3a as shown below, Fig. R28.

Fig. R28 Schematic of the strain-mediated spintronic RC system. 2 magnetic states are shown in the bottom panel and each of them can be treated as an independent physical reservoir.

Compared to other physical reservoir, the most extraordinary fact is that our RC can operate with ultra-low-power, benefiting from the strain-controlled nature. The power consumption of our

system is 2 orders of magnitude lower than the reported ones [*Nature* **547**, 428 (2017)], see Table 2 shown below.

Table 2 | Estimated power consumption for implementing the reservoir using different systems.

Reservoir type	Energy/Input	Reference
CPU-based	54.8 μ J	Nat. Electron. 2 , 480 (2019).
FPGA-based	143 nJ	
Memristor-based	3.0 nJ	
Spin-torque nano-oscillator/MTJ-based	\sim 100 fJ (1 μ W)	Nature 547 , 428 (2017).
Strain-mediated voltage-controlled super-paramagnet-based	24 fJ	Appl. Phys. Lett. 118 , 202402 (2021)
Strain-mediated Skyrmion-based	0.85 fJ	★This work

Besides ultra-low-power, the input and output signals are *all electrical* that is beneficial for real application. Recent experimental demonstrations of skyrmion-based RC using optical imaging as output [*Nat. Commun.* **13**, 6982 (2022)], and magnetic fields as inputs [*Sci. Adv.* **8**, eabq5652 (2022); Preprint at <https://arxiv.org/abs/2209.06962> (2022)]. Instead, our strain-mediated Hall bar device using *E*-field as inputs and AHE response as outputs are all electrical voltage signals, which is a step forward for physical RC applications.

And strain is a universal way to control various characteristics (see the introduction part of the revised manuscript “(lines 71-73) *Moreover, strain is a universal way to control various characteristics, which have been widely studied, including magnetization [43-48], resistance [49,50], Dzyaloshinskii–Moriya interaction (DMI) [51], phase transitions [52], luminescence [53], etc.*”), multiferroic heterostructures inherently capable for multi-parameter fusion are promising to be a powerful physical reservoir for different tasks. The strain-mediated magnetic system fits the nonlinearity naturally, which is a key precondition for RC, as shown in Fig. 2c.

Referee’s Comment #11:

(11) On line 197, please quantify "a small amount of sample data".

Authors’ Reply:

We would like to thank the referee for pointing out the unclear description. We quantify it as “The number of training data points $k = 10,000$ and validation data points of 5,000 are used for the skyrmion-enhanced strain-mediated RC system.”, which can be found in the supplementary information S6 of the revised version.

Referee’s Comment #12:

(12) In the conclusion, the authors commented the operation frequency can reach GHz. Please include any supporting data.

Authors' Reply:

We apologize for this misleading information. First of all, we need to clarify that we have not performed the high frequency operation in our reservoir. We make a prospect on its potential to work at GHz based on these reasons: First, the piezoelectric switching and ferroelectric polarization switching time can be extremely fast and less than 1 ns [*Appl. Phys. Lett.* **89**, 021109 (2006); *IEEE Int. Electron Devices Meeting (IEDM)* 15.2.1 (2019)]. Second, ferrimagnetic systems exhibit ultrafast magnetization dynamics compared to ferromagnetic system, which can reach sub-THz [*Phys. Rev. B* **100**, 100409(R) (2019)]. Third, our reservoir operates with magnetic skyrmions, where the breathing mode frequency of skyrmion is in the order of GHz. For example, Chen, X., et al. studied the characteristic frequency of the skyrmion breathing can reach above 7 GHz [*Phys. Rev. Appl.* **12**, 024008 (2019)]. With the adaption of the breathing of skyrmions, the frequency of this skyrmion based RC can be increased, and the recent work by Rajib, M.M., et al. is a good example [*Neuromorph. Comput. Eng.* **2**, 044011 (2022)]. Recently, a task-adaptive approach has been researched in a chiral magnet Cu_2OSeO_3 that host different magnetic phases including skyrmion, helical and conical, which utilizes the GHz spin dynamics of the chiral magnet [Preprint at <https://arxiv.org/abs/2209.06962> (2022)]. Those high frequency properties indicate the potential for ultrafast RC application based on ferrimagnetic skyrmions.

We have revised the manuscript with an additional *discussion* part including the potential of the strain-mediated spintronic reservoir to work at gigahertz (lines 346-352).

Referee's Comment #13:

(13) Some skyrmions in Figs. S2f and g seem to be deformed. Have the authors confirmed if these are really skyrmions (not maze domains and/or magnetic bubbles)? Please also add comments if they can treat such deformed skyrmions in the same way as the circular ones for their reservoir computation.

Authors' Reply:

We gratefully thank the referee for these instructive comments. Actually, in our RC system, as a result of strain, the elongation and recovery of skyrmions play a key role in the variation of $M_z(E)$.

For the elongated skyrmions shown in our MFM images, we can confirm that they are magnetic skyrmions. Because the skyrmions are elongated while applying the E -field, and then turned back to the regular round shape when reducing the E -field to 0 kV/cm, as shown in Fig. R29. The skyrmion winding number is unchanged due to its topological protection [*Nat. Phys.* **13**, 112 (2017); *ACS Nano* **14**, 3251 (2020)]. In the waveform classification task, the input signal (E -field) is sweeping between +0 to +15 kV/cm repeatedly, corresponding to the phase change between circular skyrmion state and deformed skyrmion state shown in Fig. R29. $\Delta M_z(E)$ of our RC system attributes to the skyrmion deformation. Similar phenomena have been reported by Ba, Y., et al. [*Nat. Commun.* **12**, 322 (2021)], where the deformation and recovery of magnetic skyrmions under E -field is shown in Fig. R30.

To avoid misleading the reader, we have revised the discussion in the manuscript as “(lines

286-290) A positive E -field of $+5\text{ kV/cm}$ is then applied (Fig. 5b). Evidently, a large amount of skyrmions is created, and some elongated shapes are observed. After reducing the E -field back to $+0\text{ kV/cm}$, the quantity of skyrmions is almost unchanged (note that the skyrmion winding number is unchanged due to its topological protection), while some elongated skyrmions restore to round shapes, as shown in Fig. 5c.”

Fig. R29 MFM images at different E -fields. $[\text{Pt/Co/Gd}]_7$ at skyrmion state. The MFM image size is $5 \times 5\ \mu\text{m}^2$.

Fig. R30 Skyrmion deformation under the E -field. [*Nat. Commun.* **12**, 322 (2021)]

Finally, we would like to thank all the referees again for the valuable suggestions and comments.
We hope the revised manuscript is now appropriate for publication in *Nature Communications*.

REVIEWERS' COMMENTS

Reviewer #1 (Remarks to the Author):

The authors have addressed my comments carefully with adequate additional experiments and analyses. I recommend the publication of the revised manuscript in Nature Communications.

Reviewer #2 (Remarks to the Author):

No further comments. I am happy with the detailed response given to the questions raised in the last round and think the work is ready to be published.

Reviewer #3 (Remarks to the Author):

(1) Regarding the first point in my initial review: the authors have determined the parity check capacity and short-term memory content, and provided the values in the response letter. However:

- a. This information should be reported in the main text of the manuscript.
- b. The reported values of roughly 2.5 are lower than alternative reservoirs, demonstrating that the proposed reservoir functions relatively poorly.

(2) Regarding the second point in my initial review: the authors have experimentally demonstrated Mackey-Glass time series prediction task, which adds significantly to the manuscript.

(3) Regarding the third point in my initial review: the authors have added a bit more detail to their energy estimate. However:

- a. The authors' assumptions that the energy requirements will scale proportionally with the area (a factor of 150,000x) and that the speed will increase by a factor of 1,000,000,000 are unreasonably optimistic.
- b. The authors do not consider the enormous energy costs for converting time-delayed sequential voltage readouts into an array of parallel data that represents the reservoir state. The high speed will likely require analog circuits (rather than digital due the high energy costs and limited speed of ADCs) with precise sample-and-hold circuits and power-hungry amplifiers to supply current to the output layer without disturbing the readout voltages. Any time-delay reservoir will therefore incur significant energy costs and severely constrain the speed of the reservoir below the GHz range, drastically limiting the utility of the proposed skyrmion reservoir.

Reviewer #4 (Remarks to the Author):

The revision made by the authors clarified all the concerns raised by the reviewer 4. This paper can now be published in Nat. Commun.

Manuscript ID: NCOMMS-22-35045A

Manuscript title: Experimental Demonstration of a Skyrmion-Enhanced Strain-mediated Physical Reservoir Computing System

Authors: Yiming Sun, Tao Lin, Na Lei, Xing Chen, Wang Kang, Zhiyuan Zhao, Dahai Wei, Chao Chen, Simin Pang, Linglong Hu, Liu Yang, Enxuan Dong, Li Zhao, Lei Liu, Zhe Yuan, Aladin Ullrich, Christian H Back, Jun Zhang, Dong Pan, Jianhua Zhao, Ming Feng, Albert Fert, and Weisheng Zhao

Response to the Report of Reviewer #1 - NCOMMS-22-35045A/Sun

Reviewer's Comment:

The authors have addressed my comments carefully with adequate additional experiments and analyses. I recommend the publication of the revised manuscript in Nature Communications.

Authors' Reply:

We sincerely appreciate the reviewer for recommending our manuscript to be published in *Nature Communications*.

Response to the Report of Reviewer #2 - NCOMMS-22-35045A/Sun

Reviewer's Comment:

No further comments. I am happy with the detailed response given to the questions raised in the last round and think the work is ready to be published.

Authors' Reply:

We sincerely grateful for the reviewer to recommend our manuscript to be published in *Nature Communications*.

Response to the Report of Reviewer #3 - NCOMMS-22-35045A/Sun

Reviewer's Comment #1:

Regarding the first point in my initial review: the authors have determined the parity check

capacity and short-term memory content, and provided the values in the response letter. However:

a. This information should be reported in the main text of the manuscript.

b. The reported values of roughly 2.5 are lower than alternative reservoirs, demonstrating that the proposed reservoir functions relatively poorly.

Authors' Reply:

We sincerely appreciate the reviewer for the comment.

a. We have added information of the parity check capacity (C_{PC}) and short-term memory content (C_{STM}) in the main text of the revised manuscript, as “(lines 199-205) *The reservoir’s computational capabilities are evaluated by further analyzing the experimental data of the waveform classification task [61,62], where $T_{delay, max} = 2$ is used in the calculation since the experimental results of the waveform classification only comprise the current, the last, and the second-to-last waveform. The parity check capacity (C_{PC}) and short-term memory content (C_{STM}) at the skyrmion state are both around 2.31, which are restricted by the slow operation of the present configuration. However, they could be significantly enhanced through involving the fast magnetization dynamics [26].*”.

b. We agree with the reviewer’s comment that the calculated C_{PC} and C_{STM} values are lower than alternative reservoirs [*Appl. Phys. Lett.* **114**, 164101 (2019); *Appl. Phys. Lett.* **121**, 102402 (2022); *Neuromorph. Comput. Eng.* **2**, 044011 (2022)]. The PC and STM tasks are used for characterizing the type of nonlinearity and the memory effect in the system, respectively. In our demonstration, we employ a prototype structure with a large Hall bar device, and the operation time (a few seconds for each waveform) is relatively long due to the limitations of the measurement setups. As a result, the memory effect and nonlinearity originated from the skyrmion dynamics (e.g. nonlinear breathing dynamics) are lost, since they occur in magnetization precession states in ns. We can expect that the memory effect and nonlinearity would be significantly enhanced through the fast magnetization dynamics in this kind of strain-mediated skyrmion RC systems, as indicated in the theoretical work where the capacity can reach $C_{STM} = 4.39$ and $C_{PC} = 4.62$ [*Neuromorph. Comput. Eng.* **2**, 044011 (2022)].

Reviewer’s Comment #2:

Regarding the second point in my initial review: the authors have experimentally demonstrated Mackey-Glass time series prediction task, which adds significantly to the manuscript.

Authors' Reply:

We sincerely thank the reviewer for this positive comment.

Reviewer’s Comment #3:

Regarding the third point in my initial review: the authors have added a bit more detail to their energy estimate. However:

a. *The authors’ assumptions that the energy requirements will scale proportionally with the*

area (a factor of 150,000x) and that the speed will increase by a factor of 1,000,000,000 are unreasonably optimistic.

b. The authors do not consider the enormous energy costs for converting time-delayed sequential voltage readouts into an array of parallel data that represents the reservoir state. The high speed will likely require analog circuits (rather than digital due the high energy costs and limited speed of ADCs) with precise sample-and-hold circuits and power-hungry amplifiers to supply current to the output layer without disturbing the readout voltages. Any time-delay reservoir will therefore incur significant energy costs and severely constrain the speed of the reservoir below the GHz range, drastically limiting the utility of the proposed skyrmion reservoir.

Authors' Reply:

We gratefully thank the reviewer for this constructive suggestion.

a. We studied the prototype of strain-mediated RC system with large Hall bar device (10 $\mu\text{m} \times 150 \mu\text{m}$) combined with PMN-PT single crystal substrate to demonstrate its functionality and physical insights. The system should be scaled down for further practical applications. For the estimation of total energy dissipation, we have chosen a 100 nm diameter skyrmion-ferroelectric hybrid disc with 100 nm thick PMN-PT, which can be experimentally realized. Firstly, skyrmions with diameter of 10 nm have been reported to be stabilized at room temperature [*Nat. Nanotech.* **13**, 1154 (2018); *Nat. Commun.* **10**, 3823 (2019)]. Thus, the 100 nm diameter disc holding several tens of skyrmions can ensure a sufficient quantity of skyrmions to possess nonlinearity, complexity and short-term memory properties for RC system. Secondly, the high quality ferroelectric PMN-PT layers still retain good ferroelectric properties as thin as 100 nm [*Crit. Rev. Solid State Mat. Sci.* **32**, 111 (2007); *Science* **334**, 958 (2011)].

The operation energy dissipated in the system is $E_s = \frac{1}{2}CV^2$, where C is the capacitance of the ferroelectric PMN-PT layer ($C = \frac{\epsilon S}{4\pi kd}$) and the V stands for voltage amplitude ($V = Ed$) [*J. Phys. D: Appl. Phys.* **44**, 265001 (2011)]. The calculated energy dissipation E_s is proportional to the area S and the PMN-PT film thickness d . When scaling down the piezoelectric substrate to a disk with 100 nm for both diameter and thickness, the energy will reduce by a factor of 4.5×10^8 as $(300 \mu\text{m} / 100 \text{nm}) \times [(10 \mu\text{m} \times 150 \mu\text{m}) / (100 \text{nm})^2]$ compared to the current prototype device.

We make a prospect on its potential to work at GHz (“the speed will increase by a factor of 1,000,000,000”) based on the following reasons: Firstly, the ferroelectric polarization switching time can be extremely fast and less than 1 ns [*Appl. Phys. Lett.* **89**, 021109 (2006); *IEEE Int. Electron Devices Meeting (IEDM)* 15.2.1 (2019)]. Secondly, ferrimagnetic systems exhibit ultrafast magnetization dynamics compared to ferromagnetic system, which can reach sub-THz [*Phys. Rev. B* **100**, 100409(R) (2019)]. Thirdly, the skyrmion dynamic frequency such as the breathing mode is in the order of GHz. We have also studied the characteristic frequency of the skyrmion breathing at about 7 GHz [*Phys. Rev. Appl.* **12**, 024008 (2019)]. With the adaption of the breathing skyrmions, the frequency of this skyrmion based RC can be increased, and the recent work by Rajib, M.M., et al. shows a good demonstration [*Neuromorph. Comput. Eng.* **2**, 044011 (2022)]. Recently, a task-adaptive approach has been studied in a chiral magnet Cu_2OSeO_3 that host different magnetic phases including skyrmion, helical and conical, which works in GHz spin dynamics of the chiral magnet

[Preprint at <https://arxiv.org/abs/2209.06962> (2022)]. Those high frequency properties indicate the potential for ultrafast operation of ferrimagnetic skyrmions based RC.

Considering the 50 MHz frequency input signal [*Neuromorph. Comput. Eng.* **2**, 044011 (2022)], the energy dissipation estimation is 340 fJ, which is the same magnitude of the previous simulation work (50 fJ) of a skyrmion-based reservoir.

b. We agree with the reviewer, we did not consider the energy cost and speed limitations for converting the voltage readouts into reservoir outputs in our previous version of the manuscript. The high energy cost and limited speed of ADCs are common problems in all physical reservoir computing systems, even in Si-based chips. Actually, we have been working on reducing the energy of ADCs for several years. We have proposed a design to convert digital input-activation into spike pulse sequence based on a novel DAC-less pulse truncation unit and employ an ADC-less charge-reservoir-integrate counter for result output [*IEEE International Conference on Integrated Circuits, Technologies and Applications (ICTA)*, 123 (2021), Best Paper Award], in which the energy cost can be reduced by 70% with a working frequency of 10 MHz, and can reach up to 500 MHz with further design optimization. Such a design can also be integrated to further reduce the energy cost in our ferrimagnetic skyrmions based physical reservoir computing systems.

Response to the Report of Reviewer #4 - NCOMMS-22-35045A/Sun

Reviewer's Comment:

The revision made by the authors clarified all the concerns raised by the reviewer 4. This paper can now be published in Nat. Commun.

Authors' Reply:

We sincerely appreciate the reviewer for recommending our manuscript to be published in *Nature Communications*.

- Finally, we would like to thank all the reviewers again for the valuable suggestions and comments, which helped us a lot to improve the quality of the manuscript. We are confident that the revised manuscript is now appropriate for publication in *Nature Communications*.